# WEAK CORRELATIONS AS THE UNDERLYING PRINCIPLE FOR LINEARIZATION OF GRADIENT-BASED LEARNING SYSTEMS

## ABSTRACT

Deep learning models, such as wide neural networks, can be conceptualized as nonlinear dynamical physical systems characterized by a multitude of interacting degrees of freedom. Such systems in the infinite limit, tend to exhibit simplified dynamics. This paper delves into gradient descent-based learning algorithms, that display a linear structure in their parameter dynamics, reminiscent of the neural tangent kernel. We establish this apparent linearity arises due to weak correlations between the first and higher-order derivatives of the hypothesis function, concerning the parameters, taken around their initial values. This insight suggests that these weak correlations could be the underlying reason for the observed linearization in such systems. As a case in point, we showcase this weak correlations structure within neural networks in the large width limit. Exploiting the relationship between linearity and weak correlations, we derive a bound on deviations from linearity observed during the training trajectory of stochastic gradient descent. To facilitate our proof, we introduce a novel method to bound the asymptotic behavior of random tensors and establish that every tensor of this kind posses a unique, tight bound.

## 1 INTRODUCTION

Deep learning in general, and particularly over-parameterized neural networks, revolutionized various fields (Graves et al. (2013); He et al. (2016); Krizhevsky et al. (2012); Silver et al. (2016)), and they are likely to do much more. Yet, the underlying reason for their unprecedented success remains elusive. These systems can be interpreted as non-linear dynamical physical systems, characterized by a multitude of interacting degrees of freedom, which makes an exact description of their behavior exceedingly hard. However, it is well established that dynamical physical systems when expanded to an infinite number of degrees of freedom tend to exhibit a simplified form of dynamics (Anderson (1972)), therefore, it seems plausible to consider such a limit in the context of deep learning systems.

A seminal study in 2018 (Jacot et al. (2018)), demonstrated that wide, fully connected neural networks, undergoing deterministic gradient descent, behave as though they were linear with respect to their parameters, (while maintaining a highly non-linear structure in their inputs). This structure has been denoted as the neural tangent kernel (NTK). The result sparked a plethora of subsequent research, generalizing it to other architectures, investigating the rate of convergence towards this linear limit, exploring the deviation of the parameters themselves from their initial configuration, decoding the structure of the kernels, and leveraging this knowledge to enhance our understanding of wide neural networks in general (Lee et al. (2019); Li et al. (2019); Cao & Gu (2019); Karniadakis et al. (2021); Huang et al. (2021); Bartlett et al. (2021)).

Subsequent discussions arose regarding the role of this limit in the exemplary performance of wide neural networks. Several studies have demonstrated that in certain contexts, infinitely wide neural networks converge to their global minimum at an exponential rate (Jacot et al. (2018); Lee et al. (2019); Du et al. (2019); Allen-Zhu et al. (2019a;b); Daniely (2017); Li & Liang (2018); Du et al. (2018); Xu et al. (2020)). Moreover, wide neural networks have been posited as effective tools for generalization, with connections drawn to the double descent phenomenon (Belkin et al. (2019); Nakkiran et al. (2021); Mei & Montanari (2022)). However, these conclusions encounter some

contention when juxtaposed with empirical evidence. Notably, several experiments indicate that for real-world data, NTK-based learning is less effective than its wide (albeit finite) neural network counterparts (Lee et al. (2020); Fort et al. (2020)). This apparent "*NTK inferiority paradox*" suggests that the relationship between the NTK limit and the success of finite neural networks may be more intricate than initially presumed.

An under-explored area within the realm of the neural tangent kernel limit pertains to the foundational principles responsible for this linearization. Chizat et al. (2019) proposed that any learning system, under a gradient-based algorithm, embodies an intrinsic scale that directs the system's linearization. Furthermore, the introduction of an external parameter, can modulate this inherent scale, thereby influencing the system's tendency towards linearization. Liu et al. (2020) demonstrated that the related ratio between the subordinate/spectral norm of the Hessian, and the euclidean norm of the Gradient that governs linearization. Their work also elucidated that in wide neural networks this ratio tends to be small.

## 1.1 OUR CONTRIBUTIONS

1. We establish that for gradient descent-based learning, linearity is equivalent to weak correlations between the first and subsequent derivatives of the hypothesis function concerning its parameters at their initial values (3.3). This equivalence is suggested as the fundamental cause for the linearization observed in wide neural networks.

2. We prove a directly that wide neural networks display this weak derivative correlations structure. By relying and extending the tensor programs formalism (Yang & Littwin (2021)), our approach uniformly addresses a broader spectrum of architectures at once, than any other proof we are aware of (4.2).

3. Drawing from the same concepts, we demonstrate how modifications in the architecture of linearizing learning systems, and more specifically, wide neural networks, affect the rate of linearization. This finding is juxtaposed Chizat et al. (2019)'s result, regarding the implications of the introduction of an external scale (3.3.2,4.2).

4. Harnessing the formalism of weak derivatives correlations, we derive a bound on the deviation from linearization over time during learning, when utilizing stochastic gradient descent (4.1). This is a generalisation of the traditional result for deterministic gradient descent (Lee et al. (2019)). This is crucial, as in most practical scenarios, stochastic gradient generalize better than deterministic gradient descent (Lee et al. (2020); Fort et al. (2020)).

5. We introduce the notion of *random tensor asymptotic behavior*, as an effective analytical tool to describe the asymptotic behavior of random tensors (2). Such tensors are not only integral to machine learning, but also serve a pivotal role in diverse mathematical and physical frameworks. Understanding the evolution of these tensors typical asymptotic behavior is relevant for addressing many questions across these fields.

The overarching simplicity and broad applicability of our findings suggest that weak derivatives correlations could very well be the foundational cause for the prevalent linearization attributes observed in wide neural networks, and possibly for other linearizing systems.

## 2 RANDOM TENSOR ASYMPTOTIC BEHAVIOR

Random tensors play a fundamental role in machine learning in general, and in this work in particular. In this section, we demonstrate the effectiveness of employing the stochastic big $O$ notation of the subordinate norm to characterize the *asymptotic behavior* of a general random tensor series (hereinafter referred to as a random tensor). Addressing the asymptotic behavior of such tensors involves two inherent challenges: the complexity arising from their multitude of components, and the stochastic nature of these components.

1. To avoid the first challenge, we focus on the tensor's subordinate norm, as elucidated in Section 2.1. This norm exhibits a wide array of useful properties, rendering it highly effective for working with random tensors, particularly concerning linear products.

2. In Section 2.2, we argue that the stochastic big $O$ notation is the most effective tool for characterizing the asymptotic behavior of random variables, surpassing other measures such as the standard big $O$ notation of the variance. We further demonstrate how its properties seamlessly synergize with those of the subordinate norm.

3. Finally, in Section 2.3, we establish that every random variable possesses a unique, tight asymptotic bound, termed the *definite asymptotic bound*. This allows us to use the stochastic big $O$ notation not merely as a tool to bound the asymptotic behavior of random tensors, but also as a way to fully characterize it. Consequently, we define the *asymptotic behavior* of a tensor $M$ as the definite asymptotic bound of the tensor's subordinate norm.

## 2.1 THE SUBORDINATE TENSOR NORM

Let $M$ be a tensor of rank $r \in \mathbb{N}_0$. Denote all its indices using the vector $\vec{i}$, such that each $i_e$ for $e = 1...r$ can assume values $i_e = 1...N_e$. Consequently, the tensor comprises a total of $N = N_1 \cdots N_r$ elements.

We will use the *subordinate norm*, defined as Kreyszig (1991):

$$\|M\| = \sup \left\{ M \cdot \left(v^1 \times \ldots \times v^r\right) \middle| v^1 \in S_{N_1} \ldots v^r \in S_{N_r}\right\} =$$
$$\sup \left\{ \sum_{i_1 \ldots i_r = 1}^{N_1 \ldots N_r} \left(M_{i_1 \ldots i_r} v^1_{i_1} \cdots v^r_{i_r}\right) \middle| v^1 \in S_{N_1} \ldots v^r \in S_{N_r}\right\} , \tag{1}$$

where $S_{N_k} = \left\{v \in \mathbb{R}^{N_k} : v \cdot v = 1\right\}$ represents the unit vectors of the appropriate dimensions. This norm satisfies certain algebraic properties outlined in lemma A.1, including: [i] the triangle inequality; [ii] for a tensor $M$ and vectors $v_1 \ldots v_q$ with appropriately defined product, the condition $\left\|M \cdot \left(v^1 \times \ldots \times v^r\right)\right\| \leq \|M\| \left\|v^1\right\| \cdots \|v^r\|$ holds; [iii] Given two tensors $M^{(1)}_{\vec{i_1}}, M^{(2)}_{\vec{i_2}}$ defining $M_{\vec{i_1}, \vec{i_2}} = M^{(1)}_{\vec{i_1}} M^{(2)}_{\vec{i_2}}$ then, $\|M\| = \left\|M^{(1)}\right\| \left\|M^{(2)}\right\|$.

Also, one has $\|M\| \leq \|M\|_F$ (with equality for vectors) (A.2) where the Frobenius norm is:

$$\|M\|_F^2 = \sum_{\vec{i}} M_{\vec{i}}^2 . \tag{2}$$

## 2.2 EFFECTIVENESS OF THE STOCHASTIC "BIG O" NOTATION

Consider a general random tensor series, denoted by $M \equiv \{M_n\}_{n=1}^{\infty}$, which henceforth we will consider as a random tensor that depend on a limiting parameter $n \in \mathbb{N}$[1].

Our objective in this section is to identify a method to describe and bound the asymptotic behavior of such a tensor, which adheres to elementary algebraic properties. Specifically, we aim for the product of multiple bounded random tensors to be constrained by the product of their respective bounds.

Employing our defined norm (1), we can simplify our problem from general random tensors to positive random variables (rank zero tensors), as our norm satisfies the elementary algebraic properties established in Lemma A.1. This reduction is substantial; however, the challenge of addressing the non-deterministic nature of our variable remains.

One might initially consider the expectation value of the tensor's norm as a solution. This approach, unfortunately falls short, because that for two positive random variables $M_1, M_2$ their product variance is not bounded by the product of their variance. In fact, generally, the converse is true:

$$\text{Var}\left(M_1 M_2\right) \geq \text{Var}\left(M_2\right) \text{Var}\left(M_1\right) \tag{3}$$

This issue becomes more pronounced when considering the product of multiple such variables, a frequent occurrence in this work. For instance, even with a basic zero-mean normal distribution with standard deviation $\sigma$, the higher moments of this distribution factor as $p!! = p(p-2)(p-4)\cdots$:

$$\forall p \in \mathbb{N} : \langle M^p \rangle = p!!\sigma^p . \tag{4}$$

---

[1]The results are applicable not only for $\mathbb{N}$, but for any other set possessing an absolute order above it

When multiplying multiple such variables, these factors can accumulate in the lower moments, rendering this definition impractical for our purposes. Similarly, any attempt to define asymptotic behavior using the variable's moments will encounter similar difficulties.

To circumvent these challenges, we adopt the stochastic big $O$ notation Dodge (2003); Bishop et al. (2007)[2]. We denote $\mathcal{N} = \left\{ f : \mathbb{N} \to \mathbb{R}^{0+} \right\}$ as the set of all functions from $\mathbb{N}$ to $\mathbb{R}^{0+}$.

**Definition 2.1** (Asymptotic Upper Bound of Random Tensors). A random tensor $M$, as defined above, is said to be asymptotically upper bounded by $f \in \mathcal{N}$ as follows:

$$M = O\left(f\right) , \tag{5}$$

if and only if:

$$\forall g \in \mathcal{N} \; s.t \; f = o\left(g\right) : \lim_{n \to \infty} P\left(\|M_n\| \leq g\left(n\right)\right) = 1 . \tag{6}$$

The lower asymptotic bound, $f = \Omega(M)$, is defined analogously but with the inequality reversed and $g = o(f)$.

Like with an infinite number of deterministic series, where pointwise convergence often falls short and uniform convergence is required, we demand a definition of a uniform asymptotic bound for discussing an infinite number of random tensors. This concept is rigorously defined in appendix A.1.

**Remark 2.1.** For a finite number of tensors, it can simply be demonstrated that the uniform bound aligns with the pointwise asymptotic bound, analogous to series convergence..

We demonstrate in lemma A.6 that this notation inherits many of the norm's properties it as defined above, including all of the properties of the subordinate norm, delineated in lemma A.1. Furthermore, it satisfies several other useful properties, outlined in appendix A.3.

### 2.3 The Definite Random Tensor Asymptotic Bound

**Remark 2.2.** We denote $f \leq g$ or $(f) \leq O(g)$ iff $f = O(g)$. We also denote $f < g$ or $O(f) < O(g)$ iff $f = O(g)$ and $f \not\sim g$, where $f \sim g \Leftrightarrow O\left(f\right) = O\left(g\right) \Leftrightarrow f = O\left(g\right) \wedge g = O\left(f\right)$. It is important to note that $f < g$ can hold even without necessitating $f = o(g)$.

It can be readily shown that for any random tensor $M$, there exist upper and lower bounds such that $O\left(h_-\right) \leq O\left(M\right) \leq O\left(h_+\right)$, and that they satisfy $h_- \leq h_+$. Furthermore, if $h_+$ and $h_-$ satisfy $h_+ \sim h_-$, their asymptotic behavior is unique. Meaning that for any other pair $h'_+, h'_-$, the relationship $h_+ \sim h'_+ \sim h'_- \sim h_-$ still holds (A.5). In such scenarios, we assert that $M$ possesses an exact asymptotic behavior, denoted as $O\left(h_+\right) = O\left(h_-\right)$.

The existence of such a pair however is not guaranteed, as illustrated by a random variable that, for every $n \in \mathbb{N}$, has equal probability of one-half to yield either $1$ or $n$. For this variable, the optimal upper bound is $n$, and the optimal lower bound is $1$, but these do not exhibit the same limiting behavior. Analogously, deterministic series may exhibit similar behavior, featuring multiple distinct partial limits. However, in the deterministic case, the *limsup* and *liminf* serve as the appropriate upper and lower limits respectively. This observation leads to the question of whether an appropriate asymptotic bound exists for the random case. It turns out, it does.

**Theorem 2.1** (Definite Asymptotic Bounds for Tensors). Consider a random tensor $M$ with a limiting parameter $n$ as described earlier. There exists $f \in \mathcal{N}$ serving as a tight/definite upper bound for $M$, satisfying:

$$M = O\left(f\right) \wedge \forall f \not< g : M \neq O\left(g\right) . \tag{7}$$

Furthermore, the asymptotic behavior of $f$ is unique.

***Explanation***. Although the theorem may appear intuitive, the challenge arises from the fact that our order above $\mathcal{N}$ is not a total one, even when considering only the asymptotic behavior of the functions. For example, none of the following equations hold true:

$$\sin\left(\pi n\right) < \cos\left(\pi n\right), \cos\left(\pi n\right) < \sin\left(\pi n\right), \sin\left(\pi n\right) \sim \cos\left(\pi n\right) . \tag{8}$$

We address this issue by employing Zorn's lemma, as demonstrated in appendix A.2. $\qquad \square$

---

[2]Our definition slightly differs from the standard definition for big $O$ in probability notation, but it is straightforward to show its equivalence

Since every such random tensor $M$ has precisely one definite asymptotic bound $f$, we can consider this bound as the *random tensor's asymptotic behavior*, represented as:

$$O(M) = O(f) . \tag{9}$$

## 3 WEAK CORRELATIONS AND LINEARIZATION

### 3.1 NOTATIONS FOR SUPERVISED LEARNING

#### 3.1.1 GENERAL NOTATIONS

Supervised learning involves learning a *classifier*: a function $\hat{y} : X \to Y$ that maps an input set (here $X \subseteq \mathbb{R}^{d_X}$), to an output set (here $Y \subseteq \mathbb{R}^{d_Y}$), given a dataset of its values $X' \subseteq X$, denoted as the "*target function*". This is achieved by using an *hypothesis function*, in our case of the form $F : \mathbb{R}^N \to \{f : X \to Y\}$ which depends on certain parameters $\theta \in \mathbb{R}^N$ (in the case of fully connected neural networks for example, the weights and biases). The objective of supervised learning is to find the optimal values for these parameters, such that $F$ captures $\hat{y}$ best, with respect to a cost function $\mathcal{C}$. We use $x \in X$ to denote elements in the input set, and $i, j = 1...d_Y$ to denote the output vector indices. The parameters $\theta$ are enumerated as $\theta_\alpha, \alpha = 1, ..., |\theta| = N$, and their initial values are denoted by $\theta_0 = \theta(0)$.

We work within the optimization framework of single input batches gradient descent-based training, which is defined such that for every learning step $s \in \mathbb{N}$:

$$\Delta^{x_s}\theta(s) = \theta(s+1) - \theta(s) = -\eta\nabla\mathcal{C}(F(\theta)(x_s), \hat{y}(x_s))|_{\theta=\theta(s)} =$$
$$= -\eta\nabla F(\theta(s))(x_s)\mathcal{C}'(F(\theta(s))(x_s), \hat{y}(x_s)) . \tag{10}$$

Here, $\nabla_\alpha = \frac{\partial}{\partial\theta_\alpha}$ represents the gradient operator, $x_s$ denotes the $s \in \mathbb{N}$th input data, and $\mathcal{C}'(x) = \frac{d\mathcal{C}(x)}{dx}$ refers to the derivative of the cost function. The derivative matrix/the Jacobian $\nabla F$ is defined such that for every indices $i, \alpha$, $(\nabla F)_{\alpha i} = \nabla_\alpha F_i$. We denote $\eta$ as the learning rate and $(x_s, \hat{y}(x_s))$ as the images and labels, respectively. The training path is defined as the sequence of inputs upon which we trained our system, represented by $\{x_s \in A\}_{s=0}^\infty$. We assume that each input along this path is drawn from the same random distribution $\mathcal{P}$, neglecting the possibility of drawing the same input multiple times. The same distribution will be used for both training and testing. Moreover, we assume that the hypothesis function and the cost function $F, \mathcal{C}$ are analytical in their parameters. We study learning in the limit where the number of parameters $N \equiv |\theta| \to \infty$, with $N \equiv N(n)$ being a function of some other parameter $n \in \mathbb{N}$, denoted as the "limiting parameter". For neural networks, $n$ is typically chosen as the width of the smallest layer, but we can choose any parameter governs the system's linearization.

**Remark 3.1.** This framework can be greatly generalised, as we discussed in appendix F.

#### 3.1.2 NEURAL TANGENT KERNEL NOTATIONS

Numerous gradient descent learning systems (GDML) with different neural network architectures, display a linear-like structure in their parameters in the large width limit. In this linear limit, the hypothesis function takes the following form:

$$F_{lin}(0) = F(\theta_0),$$
$$\forall s \in \mathbb{N}_0 : F_{lin}(s+1) = F_{lin}(s) - \Theta_0(\cdot, x_s)\mathcal{C}'(F_{lin}(s)(x_s), \hat{y}(x_s)) , \tag{11}$$

with the kernel $\Theta$ defined such as:

$$\forall x, x' \in X : \Theta(\theta)(x, x') = \eta\nabla F(\theta)(x)^T \nabla F(\theta)(x') , \quad \Theta_0 \equiv \Theta(\theta_0) , \tag{12}$$

where $\nabla F^T$ is the transpose of $\nabla F$ the Jacobian.

### 3.2 THE DERIVATIVES CORRELATIONS

#### 3.2.1 THE DERIVATIVES CORRELATIONS DEFINITION

In the following, we prove that linearization is equivalent to having weak correlations between the first, and higher derivatives of the hypothesis function, with respect to the initial parameters. We define the *derivative correlations* as follows:

**Definition 3.1** (Derivatives Correlations). We define the derivatives correlations of the hypothesis function for any positive integer $d \in \mathbb{N}$ and non-negative integer $D \in \mathbb{N}^0$ as:

$$\mathfrak{C}^{D,d}(\theta) = \frac{\eta^{\frac{D}{2}+d}}{D!d!} \nabla^{\times D+d} F(\theta)^T (\nabla F(\theta))^{\times d} , \tag{13}$$

where the higher order derivatives defined such that for every $d \in \mathbb{N}$ and indices $i, \alpha_1 \ldots \alpha_d$, $\left(\nabla^{\times D} F\right)_{\alpha_1 \ldots \alpha_d, i} = \nabla_{\alpha_1} \cdots \nabla_{\alpha_d} F_i$.

More explicitly, we present the inputs and indices of these tensors as follows:

$$\mathfrak{C}^{D,d}(\theta)_{i_0, i_1 \ldots i_d}^{\alpha_{1+d} \ldots \alpha_{D+d}}(x_0, x_1 \ldots x_d) =$$
$$\frac{\eta^{\frac{D}{2}+d}}{D!d!} \sum_{\alpha_1 \ldots \alpha_d=1}^{N} \nabla_{\alpha_1 \ldots \alpha_{D+d}}^{\times D+d} F_{i_0}(\theta)(x_0) \left(\nabla_{\alpha_1} F_{i_1}(\theta)(x_1) \cdots \nabla_{\alpha_d} F_{i_d}(\theta)(x_d)\right) , \tag{14}$$

The objects in (13) are the correlation of the derivatives in the sense that $\alpha_1 \ldots \alpha_d$ can be viewed as random variables, drawn from a uniform distribution of $\{1 \ldots N\}$, while $\theta$ and all other indices are fixed instances and hence deterministic. In this context, $\nabla^{\times D+d} F$ and $\nabla F \times \ldots \times \nabla F$ in (13) can be viewed as random vectors of the variables $\alpha_1 \ldots \alpha_d$, and the summation in (13) represents the (unnormalized) form of the "Pearson correlation" between the two random vectors. The overall coefficient of the learning rate $\eta^{\frac{D}{2}+d}$ serves as the appropriate normalization, as we will demonstrate in appendix C and D. We will also denote: $\mathfrak{C}^d(\theta) \equiv \mathfrak{C}^{0,d}(\theta), \mathfrak{C}^{D,d} \equiv \mathfrak{C}^{D,d}(\theta_0), \mathfrak{C}^d \equiv \mathfrak{C}^d(\theta_0)$.

An example for these correlations is the $D = 0, d = 1$ correlation, the correlation of the first derivative with itself, the kernel:

$$\mathfrak{C}^1(\theta) = \eta \nabla^T F(\theta) \nabla F(\theta) = \Theta(\theta) . \tag{15}$$

The definition for the asymptotic behavior for these derivative correlations is slightly nuanced due to the many different potential combinations of distinct inputs. We rigorously define it in appendix B.1.

### 3.3 EQUIVALENCE OF LINEARITY AND WEAK DERIVATIVES CORRELATIONS

Our main theorems concern the equivalence of linearity and weak derivative correlations. In other words, weak correlations can be regarded as the fundamental reason for the linear structure of wide neural networks. These theorems are applicable for systems that are properly scaled in the initial condition, meaning that when taking $n \to \infty$ the different components of the system remain finite. We define in rigour exactly what it means in appendix B.2. We denote such systems as properly normalised GDMLs or *PGDMLs*.

#### 3.3.1 OUR MAIN THEOREMS

In the following theorems, we describe two distinct manifestations of the equivalence between linearization and weak derivatives correlations for a PGDMLs. We denote by $m(n)$ as the parameter of the linearization/correlation decay where $m(n) \to \infty$. $m(n)$ is an intrinsic parameter of the system, and is defined by the linearization rate or the correlation structure. For wide neural networks for example, $m(n) = \sqrt{n}$.

**Theorem 3.1** (Fixed Weak Correlations and Linearization Equivalence). Under the conditions described above, for a sufficiently small learning rate $\eta < \eta_{the}$, the two properties are equivalent:

1. $m(n)$ - fixed weak derivatives correlation:

$$\forall d, D \in \mathbb{N} : \mathfrak{C}^d = O\left(\frac{1}{m(n)}\right), \mathfrak{C}^{D,d} = O\left(\frac{1}{\sqrt{m(n)}}\right) \quad \text{Uniformly.} \tag{16}$$

2. Simple linearity: For every fixed training step $s \in \mathbb{N}$:

$$F(\theta(s)) - F_{lin}(s) = O\left(\frac{1}{m(n)}\right) ,$$
$$\forall D \in \mathbb{N} : \eta^{\frac{D}{2}} \left(\nabla^{\times D} F(\theta(s)) - \nabla^{\times D} F(\theta_0)\right) = O\left(\frac{1}{\sqrt{m(n)}}\right) \quad \text{Uniformly.} \tag{17}$$

$\eta_{the}$ is defined such as all the correlations are uniformly bounded by $O(1)$, to ensure the sum converges, as shown in appendix C.2.

The next theorem delineates an even stronger equivalence, which is also relevant for wide neural networks. It also encompasses the scaling of the learning rate.

**Theorem 3.2** (Exponential Weak Correlations and Linearization Equivalence). For the conditions described above, the two properties are equivalent:

1. $m(n)$ - power weak derivatives correlation:

$$\forall \left( D, \in \mathbb{N}^0, d \in \mathbb{N} \right) \neq (0,1) : \mathfrak{C}^{D,d} = O \left( \frac{1}{\sqrt{m(n)}} \right)^d \quad \text{Uniformly.} \quad (18)$$

2. Strong linearity: For every reparametrisation of the learning rate $\eta \to r(n)\eta$, $r(n) > 0$ and for every fixed training step $s \in \mathbb{N}$:

$$F(\theta(s)) - F_{lin}(s) = O\left( \frac{r(n)}{m(n)} \right),$$
$$\forall D \in \mathbb{N} : \left( \frac{\eta}{r(n)} \right)^{\frac{D}{2}} \left( \nabla^{\times D} F(\theta(s)) - \nabla^{\times D} F(\theta_0) \right) = O\left( \frac{r(n)}{\sqrt{m(n)}} \right). \quad (19)$$

***Explanation***. We prove the theorems by considering for a general learning step $s \in \mathbb{N}$, the hypothesis function and its derivatives' Taylor series expansion around the $s-1$ step. Utilizing equation 10, we can find that the evolution of the derivatives of $F$ and its derivatives during learning, is governed by a linear combination of the correlations of the form:

$$\forall D \in \mathbb{N}^0 : \Delta \frac{\eta^{\frac{D}{2}}}{D!} \nabla^{\times D} F(\theta) = \sum_{d=1}^{\infty} \mathfrak{C}^{D,d}(\theta) \left( -\mathcal{C}'(F(\theta), \hat{y}) \right)^{\times d}, \quad (20)$$

where $\Delta \nabla^{\times D} F$ in the change of $\nabla^{\times D} F$. For deterministic functions it is straightforward to prove the equivalences by employing the arithmetic properties of the big $O$ notation, and that [i] One can choose any $F - \hat{y}$ (as long as its asymptotic behavior is appropriate). [ii] Different components in our sum cannot cancel each other, since we can change $\eta$ continuously; thus, for the sum stay small, all of the components must be small. The adjustments needed for our case of stochastic functions are minor, as, as we show in appendix A.3, our tensor asymptotic behavior notation satisfies many of the same properties of the deterministic big $O$ notation. The complete proofs are in appendix **??**. □

### 3.3.2 EXTERNAL SCALE AND HESSIAN SPECTRAL NORM

We see in theorem 3.2, that a rescaling of $\eta$ such as $\eta \to r(n)\eta$ can either promote or impede the process of linearization. This observation also holds for Theorem (3.2) as long as $\eta < \eta_{the}$. This insight offers a deeper understanding of the findings presented by Chizat et al. (2019). Specifically, it elucidates that an alteration of an external scale influences linearization by affecting the scale of the higher correlations differently than of the lower ones.

A notable connection to another principal research Liu et al. (2020), is the definition of derivatives correlations themselves. In Liu et al. (2020), the authors established that linearization, results from a small ratio between the spectral norm of the Hessian and the norm of the gradient. The derivative correlations can be interpreted as a spectral norm, but concerning solely the gradient, when considered as a vector. This interpretation serves as a refinement of the results presented in Liu et al. (2020). Unlike in Liu et al. (2020) approach, which required this ratio to be small within a ball, our approach demands its minimization at the initialization point itself. Because of that it necessitates the decay of higher-order correlations.

### 3.3.3 THE CHICKEN AND THE EGG OF LINEARIZATION AND WEAK CORRELATIONS

The relationship between linearization and weak correlations in over-parameterized systems can be comprehended from two different viewpoints. The first perspective suggests that effective learning

in such systems necessitates a form of implicit regularization, which inherently favors simplicity (Belkin et al. (2019)). This preference can be directly incorporated by imposing a linear (or at least approximately linear), structure in highly over-parameterized regimes. Notably, in certain scenarios, linearization can facilitate exponential convergence rates, especially with respect to the training datasets and, but in some instances, even with respect to the testing datasets (Jacot et al. (2018); Lee et al. (2019); Du et al. (2019); Allen-Zhu et al. (2019b); Daniely (2017); Li & Liang (2018); Du et al. (2018); Xu et al. (2020); Allen-Zhu et al. (2019a)). Hence, weak derivative correlations can be interpreted as a pragmatic approach for achieving linearization.

An alternative interpretation, aligning more closely with the spirit of this paper, suggests that weak derivative correlations do not primarily serve as a dynamic mechanism for linearization, but rather, as its underlying cause. In this context, persisting derivative correlations may indicate an inherent bias within the system, typically undesirable. Therefore, linearization can be viewed as a consequence of our attempt to avoid counterproductive biases, by demanding weak correlations.

Moreover, if we possess some prior knowledge about an inherent biases in our problem, it might be advantageous to allow some non-decaying correlations, counteracting the process of linearization. Furthermore, as certain biases can enhance general learning algorithms (in the form of implicit and explicit regularization), this perspective might provide valuable insights into the "NTK inferiority paradox" introduced in the introduction (1). The reason why linear learning underperforms in comparison to finite neural networks, might be that it lack some beneficial biases, in the form of non vanishing correlations.

## 4 PROPERTIES OF WEAKLY CORRELATED PGDMLS

### 4.1 APPLICATION: DEVIATION FROM LINEARITY DURING LEARNING

Multiple studies have examined the deviation of the hypothesis function $F$ from its linear approximation, $F_{lin}$ (11), as a function of $n$ for a fixed learning step (especially in the context of wide neural networks). Yet, it seems that no research has explored the deviation between these functions with respect to the learning step for stochastic GD (10). This aspect is crucial since even if $F - F_{lin}$ vanished for a given learning step, if it deviates too fast during learning, the linearization may not be evident for realistic large $n$.

We address single-input batches stochastic GD in our study. However, as we explained in appendix F, this result can be greatly generalized. Notably, the analysis for stochastic GD may be even more relevant even for deterministic GD, than the conventional approaches that presuppose a training dataset. This is because, while the batch might be fixed, its initial selection is from a stochastic distribution.

**Corollary 4.0.1** (Weakly Correlated PGDML Deviation Over Time). For an exponentially $m(n)$-weakly correlated PGDML, given $\eta < \eta_{cor}$, and some $S \in \mathbb{N}$, that for every $s = 1 \ldots S$, if:

$$\mathcal{C}'\left(F_{lin}\left(s\right), \hat{y}\right) = O\left(e^{-\frac{s}{T}}\right), \mathcal{C}''\left(F_{lin}\left(s\right), \hat{y}\right) = O\left(1\right) \quad \text{Uniformly,} \tag{21}$$

than:

$$F\left(\theta\left(s\right)\right) - F_{lin}\left(s\right) = O\left(\frac{s^0}{m\left(n\right)}\right) \quad \text{Uniformly.} \tag{22}$$

$\eta_{cor}$ is the standard critical learning rate ensuring our system's effectively learns in the NTK limit (D). It's typically from the same order of magnitude as $\eta_{the}$.

***Explanation***. We prove the corollary by using a similar induction process as in theorems (3.1,3.2). However, here we also consider the dependency in the learning step, as detailed in appendix D. We are able to bound the deviation over time, by leveraging the fact that in the NTK limit during the initial phases of the learning process, the system converges towards the target function exponentially fast[3] (Jacot et al. (2018); Lee et al. (2019); Du et al. (2019); Allen-Zhu et al. (2019b); Daniely (2017); Li & Liang (2018); Du et al. (2018); Xu et al. (2020); Allen-Zhu et al. (2019a)). We believe that subsequent research will be able to produce more refined bounds. □

---

[3]The known bounds for $\mathcal{C}'\left(F_{lin}, \hat{y}\right)$ are typically bounds over the variance. In appendix A.4, we discuss how an average exponential bound can be translated into a uniform probabilistic bound.

## 4.2 EXAMPLE: WIDE NEURAL NETWORKS

Numerous studies have demonstrated that a wide range of neural networks architectures exhibit linearization as they approach the infinite width limit, including any combination of CNNs, convolutional neural network, recurrent neural networks, attention, and others. However, the existing proofs tend to be specific to particular architectures, and are often intricate in nature. The most comprehensive proof we aware of that uniformly encompasses a diverse set of architectures, is presented in (Yang & Littwin (2021); Yang (2020)). These works employed the tensor product formalism (Yang (2019)), which can describe most relevant variants of wide neural network architectures, as the composition of global linear operations, and point-wise non linear functions.

1. Relying on the semi-linear structure of FNCs we were able show explicitly by induction that for appropriate activation functions wide neural networks are $n$-fixed weakly correlated, and $n^{\frac{3}{2}}$-exponential weakly correlated, (and in most practical seance can be considered as $n$-exponentially weakly correlated as well), (E).

2. The framework of low correlations proves effective in discerning how modifications to our network influence its linearization. For instance, it is evident that $\sup_{n \in \mathbb{N}} \frac{\phi^{[n]}}{(n+1)!}$, govern the rate of linearization in FNCs (E). This observation is why we demand for FNCs, that over the relevant domain, the activation function satisfy:

$$\phi^{[n]} \leq O\left((n+1)!\right) , \tag{23}$$

where $\phi^{[n]}$ is the $n$-th derivative of the network's activation function - $\phi$.

3. Our proof for FNCs can simply be generalised for any wide network, described by the tensor programs formalism (E.5.1). This is because, similarly to FNCs, all such systems exhibit a wide semi-linear form by definition. Demonstrating that the linearization of these systems arises from weak correlations, allows us to utilize all of the insights we've found for weakly correlated systems in general. We were also been able to conceive linearizing network-based systems, that fall outside the scope of the tensor programs formalism (E.5.2).

   Leveraging the notation of the asymptotic tensor behavior, our proof accommodates a broad spectrum of initialization schemes, extending beyond the Gaussian initialization predominantly employed in other studies.

## 5 DISCUSSION AND OUTLOOK

The linearization of large and complex learning systems is a widespread phenomenon, but our comprehension of it remains limited. We propose the weak derivatives correlations (3.1), is the underlying structure behind this phenomenon. We demonstrated that this formalism is natural for analyzing this linearization: [i] It allows for the determination of if, and how fast a general system undergoes linearization (3.3.1,4.2). [ii] It aids us in analyzing the deviations from linearization during learning (4.0.1).

These insights raise a pivotal question (discussed in 3.3.3): Is the emergence of the weak correlations structure simply a tool to ensure a linear limit for overparameterized systems? Or does weak correlations indicate an absence of inherent biases, leading to linearization? If the latter is true, it suggests that in systems with pre-existing knowledge, specific non-linear learning methodologies reflecting those biases might be beneficial. That could partially explain why the NTK limit falls short in comparison to finite neural networks.

At the core of our weak derivatives correlation framework, is the random tensor asymptotic behavior formalism, outlined in section 2. We have showcased its efficacy in characterizing the asymptotic behavior of random tensors, and we anticipate its utility to extend across disciplines that involve such tensors.

We further discuss generalisations and limitations in appendix F.

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
