## 6 COMMENTS

### 6.1 REVIEW 1

Thank you for your detailed and important review. It is very instructive.

We acknowledge the omission of Liu et al. (2020), which is indeed relevant to our work. We apologize for this oversight and have now thoroughly incorporated and referred to it in section 3.3.2.

1. Following a dissection with the authors of Yang & Littwin (2021) and have corrected the relevant text in Section 4.2 and appendix E.5.

2. A comprehensive introduction has been included at the beginning of section 2 to better elucidate the motivations behind our formalism.

3. The previously stated requirement for the cost function as $F - \hat{y}$ was indeed unnecessary and has been rectified in the current version.

4. That indeed was an error that has been corrected.

5. An extensive explanation regarding the limiting parameter has been added towards the end of section 3.1.

6. We have clarified the equation in question by explicitly stating it is relevant for the linear limit.

7. The correlations in equation 13 are in the sense of the Pearson correlation coefficient, and in the same way we can consider the indices as random parameters. We have clarified this in our revised manuscript.

8. The relevance of these correlations, now clearly defined, and their relationship to linearization is elaborated in section 3.3.1 and equation 20.

9. The $O$ referenced is the same as in Section 2.3.

10. We intended to denote $n$ to the power of zero, effectively 1.

11. Thank you for pointing out the error in equation 17, now corrected in equation 73.

12. A more intuitive and precise explanation for the applicability of our results to realistic systems is now included in B.2.

13. The parameter referred to is as defined in theorems 3.1,3.2.

14. We have relocated the discussion to appendix F to provide a more in-depth analysis. we explain that our assertions are valid for wide neural networks, as rigorously proven in appendix E, which is why we focus on scenarios where these conditions are met.

15. These systems are dynamical, as they develop over time. Furthermore, outside the NTK limit, they are non linear.

16. We focused on the transpose of the Jacobian for the definition of correlations in equation 13, to align with the standard inner product format in linear algebra. This aspect has been further clarified in the manuscript, particularly in section 3.1.1, where it is explicitly stated that the matrix referred to is the Jacobian transpose.

17. The notation has been adjusted to standard transpose format.

18. We defined it more clearly. How would you prefer for us to write that, with the $\times$? We wanted to make sure that people understand it is tensor power of the gradient.
    We have defined it more clearly in section 3.2.1. How would you suggest we reformulate the equation? without the $\times$ symbol? Our intention is to clearly convey that it represents the tensor power of the gradient.

### 6.2 REVIEW 2

Thank you for your valuable review and suggestions for improving the paper.

- Significant efforts have been made in this version to enhance readability and clarity. The contributions section (1.1) has been made more explicit, and the entire paper has been revised for better comprehension.

- Regarding the distinction between standard large $O$ notation in probability and our approach in section 2, we affirm there is no fundamental difference in terms of bounding random variables. This clarification has been emphasized in the current version. Our contribution is twofold:

  1. We established that every random variable possesses a unique asymptotic bound, allowing the use of large $O$ notation not only for bounding but also for fully characterizing the asymptotic behavior of random variables.
  2. We demonstrated the compatibility of large $O$ notation properties with those of the subordinate norm.

  These insights underpin our proposition that interpreting the asymptotic behavior of a random tensor, as the tight bound of its subordinate norm provides a novel and useful perspective for understanding random tensors' asymptotic behavior, as emphasized in the revised manuscript.

- Our findings considerably extend the scope of those by Li et al. (2019), covering a wider range of architectures. They are also more comprehensive even than the work of Yang & Littwin (2021), which already spans a broad array of architectures. Our methodology is particularly effective in analyzing linearized systems, as highlighted in the updated section 4.2.

  The revised sections 1.1,4.2 should more effectively address your questions regarding the breadth and implications of our approach relative to existing literature.

- We included detailed heuristic proofs at the outset of each appendix. This addition aims to enhance the paper's readability and comprehension. Furthermore, we improved the explanations of the theorems in the main text.

- Regarding the remark about NTK learning not converging to the target function exponentially fast: We acknowledge this except when the target function is within the kernel's conjugate class Daniely (2017). However, our reference in section 4.1 pertains to the initial stages of learning, where convergence is typically exponential.

## 6.3 REVIEW 3

Thank you for your important review, comments and suggestions.

1. Thank you for your constructive suggestions. We have revised the paper accordingly and made an effort to improve overall readability and coherence.

   $m(n)$ is an intrinsic parameter of the system, and it is defined by it. We emphasised that in our new version in the beginning of section 3.3.1. The definition of big $O$ with subscripts is given in B.1. We also defined $\nabla^{\times D} F$ explicitly in section 3.1.1.

2. A detailed discussion of the scope, limitations, and disadvantages has been included in appendix F, with references made in the discussion section (5).

3. The concept of 'properly scaled systems' is what we defined as PGDMLs. We clarified that in section 3.3 and defined it with rigour in appendix B.2.

4. Considerable effort has been invested in enhancing the clarity and readability of the paper. We trust that these revisions have significantly elevated the paper's standard.

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

## A  RANDOM TENSORS ASYMPTOTIC BEHAVIOR

In the following sections, we utilize the results of this section throughout our analyses repeatedly. Due to their intuitive nature, we may not consistently specify when we do so, and which lemma/theorem we are employing.

### A.1  PROPERTIES OF OUR NORM

In this section, we explore the properties satisfied by the subordinate norm. We omit the proofs as these properties are either well-known, or straightforward to prove, (and also enjoyable to derive).

**Lemma A.1** (Algebraic properties of the subordinate norm). The subordinate tensor norm (1) satisfies the following algebraic properties:

1. Given a tensor series $\left\{ M^{(d)} \right\}_{d=1}^{D}$ where $D \in \mathbb{N} \cup \{\infty\}$, it satisfies the triangle inequality:

$$\left\| \sum_{d=1}^{D} M^{(d)} \right\| \leq \sum_{d=1}^{D} \left\| M^{(d)} \right\| , \tag{24}$$

   where equality holds when the tensors are identical.

2. Given a tensor $M_{i_1 \dots i_r}$, $1 \leq i_k \leq N_k$ for $1 \leq k \leq r$, and $q \leq r \in \mathbb{N}$ vectors $v_{i_1}^1 \dots v_{i_q}^q$ (with the same range of indices), then:

$$\left\| M \cdot v^1 \times \cdots \times v^q \right\| \leq \| M \| \left\| v^1 \right\| \cdots \left\| v^q \right\| . \tag{25}$$

3. Given two tensors $M_{\vec{i_1}}^{(1)}$ and $M_{\vec{i_2}}^{(2)}$, their direct product $M_{\vec{i_1} \vec{i_2}} = \left( M^{(1)} * M^{(2)} \right)_{\vec{i_1} \vec{i_2}} = M_{\vec{i_1}}^{(1)} M_{\vec{i_2}}^{(2)}$, satisfies:

$$\| M \| = \left\| M^{(1)} \right\| \left\| M^{(2)} \right\| . \tag{26}$$

   The generalization an arbitrary finite number of tensors is trivial.

**Remark A.1.** Parts 1 and 3 are also satisfied by the Frobenius norm.

**Lemma A.2** (Relation to the Frobenius Norm). Given a tensor $M$ of rank $r \in \mathbb{N}$, the following holds:

1. For any tensor $M$:

$$\| M \| \leq \| M \|_F , \tag{27}$$

   and if $r = 1$ (i.e., the tensor is a vector), then:

$$\| M \| = \| M \|_F = \sqrt{\sum_i M_i^2} . \tag{28}$$

2. For every $r' = 1 \dots r$:

$$\| M \| = \sup \left\{ \left\| M \cdot \left( \begin{array}{c} v^1 \times \dots \times v^{r'-1} \times \\ v^{r'+1} \times \dots \times v^r \end{array} \right) \right\|_F \; \middle| \; \begin{array}{c} v^1 \in S_{N_1} \dots v^{r'-1} \in S_{N_{r'-1}} \\ v^{r'+1} \in S_{N_{r'+1}} \dots v^r \in S_{N_r} \end{array} \right\} . \tag{29}$$

The first part of the lemma demonstrates that our norm is always bounded by the Frobenius norm, and the two norms coincide for vectors. The second part generalizes the first, indicating that when reducing any tensor to a vector, the two norms once again agree.

**Lemma A.3** (Properties of the Maximizing Vectors). Given a tensor $M$ of rank $r \in \mathbb{N}$, there exist vectors $v^1 \dots v^r$ of norm 1 such that:

$$\| M \| = M \cdot v^1 \times \cdots \times v^r . \tag{30}$$

This result indicates that the supremum is indeed a maximum. The vectors $v^1 \dots v^{r'-1}, v^{r'+1} \dots v^r$ are also the ones that maximize the cases demonstrated in the previous lemma.

Moreover, if the tensor is symmetric with respect to the permutation of the indices $i_1, i_2, \ldots, i_q$ and is non-zero, then:

$$v^{i_1} = v^{i_2} = \cdots = v^{i_q} \ . \tag{31}$$

**Remark A.2.** For $M = 0$, any set of vectors maximizes our result, irrespective of whether the vectors are identical or distinct.

## A.2 EXISTENCE AND UNIQUENESS OF THE TENSOR ASYMPTOTIC BEHAVIOR

In this section, we discuss some of the more general properties that the tensor asymptotic behavior notation satisfies, regardless of the norm it is defined with respect to. The first lemma we present is a useful equivalent definition for bounding tensor asymptotic behavior. This equivalent definition will be beneficial for our later discussion:

**Lemma A.4** (Equivalent Definitions for Tensor's Asymptotic Bound). For any random tensor $M$ and $f \in \mathcal{N}$, the two definitions for bounding the tensor's asymptotic behavior $O(M) \leq O(f)$ are equivalent (the first is the original definition, (2.1)):

1. 

$$\forall g \in \mathcal{N} \ s.t \ f = o(g) : \lim_{n \to \infty} P\left(\|M_n\| \leq g(n)\right) = 1 \ . \tag{32}$$

2. 

$$\lim_{c \to \infty} \lim_{n \to \infty} P\left(\|M_n\| \leq cf(n)\right) = 1 \ . \tag{33}$$

(The same applies for $O(f) \leq O(M)$).

The order in which we take the limits in equation 33 is crucial, as any random tensor satisfies the equation for any $f$, if we take first the limit of $c$.

It is straightforward to show that any random tensor $M$ has lower and upper bounds:

**Lemma A.5** (Bounding Tensor's Magnitude). Given a random tensor $M$, there exist $h_-, h_+ \in \mathcal{N}$ such that:

$$O\left(h_-\right) \leq O\left(M\right) \leq O\left(h_+\right) \ . \tag{34}$$

The fact that such bounds always exist demonstrates that our notation is well-founded and natural. An even stronger evidence for this, which for reasons discussed in section 2.3 is much less simple to show, is that **there is always one well-defined "best" upper bound - Theorem 2.1**. We prove this theorem after Lemma A.5 by using Zorn's lemma.

**Remark A.3.** It is simple to show that if there exist lower and upper bounds such that $h_+ = h_-$ and the exact asymptotic behavior is well defined, they are the "definite bound" of Theorem 2.1.

*Proof - Lemma A.4.*

We will prove the two directions of the lemma separately.

Assuming the second condition in equation 33 is satisfied:

Given some $0 < p < 1$, we know using equation 33 that there is some $0 < c$ such that for sufficiently large $n \in \mathbb{N}$:

$$p \leq P\left(\|M_n\| \leq cf(n)\right) \ . \tag{35}$$

Given some $g \in \mathcal{N}$ such that $f = o(g)$, we know that for sufficiently large $n \in \mathbb{N}$:

$$cf(n) \leq g(n) \ , \tag{36}$$

which means that for sufficiently large $n \in \mathbb{N}$:

$$p \leq P\left(\|M_n\| \leq cf(n)\right) \leq P\left(\|M_n\| \leq g(n)\right) \ . \tag{37}$$

As we proved that for any $0 < p < 1$ we get that:

$$\lim_{n \to \infty} \left(P\left(\|M_n\| \leq g(n)\right)\right) = 1 \ . \tag{38}$$

And as we proved that for any arbitrary $g \in \mathcal{N}$ such that $f = o(g)$, we proved the first part of the lemma.

Assuming the first condition in equation 32, is satisfied:

If we assume in contradiction that equation 33 is not satisfied, we get that there is some $0 < p < 1$ such as:

$$\forall n_0 \in \mathbb{N} \, 0 < c \, \exists n_0 \leq n \in \mathbb{N} : P\left(\|M_n\| \leq cf(n)\right) < p \, . \tag{39}$$

In particular that means that if we choose the series $\{c_i = i\}_{i=1}^{\infty}$, there are $\tilde{n}_1 < \tilde{n}_2 < \tilde{n}_3... \in \mathbb{N}$ such as:

$$\forall i \in \mathbb{N} : P\left(\|M_{\tilde{n}_i}\| \leq if(\tilde{n}_i)\right) < p \, . \tag{40}$$

The reason that we can require that $\{\tilde{n}_i\}_{i=1}^{\infty}$ is rising, is that we know that we can find such $n$-s for any sufficiently large $n_0$ and for any $c$. So by induction we can require every time that every $\tilde{n}_i$ is bigger than all previous $\tilde{n}$-s.

Assuming the first condition of equation 32 is satisfied:

Suppose, by contradiction, that equation 33 is not satisfied. Then, there exists some $0 < p < 1$ such that:

$$\forall n_0 \in \mathbb{N}, \, 0 < c, \, \exists n_0 \leq n \in \mathbb{N} : P\left(\|M_n\| \leq cf(n)\right) < p \, . \tag{41}$$

In particular, if we choose the series $\forall i \in \mathbb{N} : c_i = i$, there exist $\tilde{n}_1 < \tilde{n}_2 < \tilde{n}_3 \cdots \in \mathbb{N}$ such that:

$$\forall i \in \mathbb{N} : P\left(\|M_{\tilde{n}_i}\| \leq if(\tilde{n}_i)\right) < p \, . \tag{42}$$

Since we can find such $n$-values for any sufficiently large $n_0$ and any $c$, and we can require by induction that each $\tilde{n}_i$ is greater than all previous $\tilde{n}$-values. .

We can now define the function:

$$\forall n \in \mathbb{N} : g(n) = \left(\min\{i \in \mathbb{N} \mid \tilde{n}_i \leq n\}\right) f(n) \, . \tag{43}$$

Since $\{\tilde{n}_i\}_{i=1}^{\infty}$ is increasing, we know by the Archimedean property that $\min\{i \in \mathbb{N} \mid \tilde{n}_i \leq n\}$ is also increasing and unbounded, which implies:

$$\lim_{n \to \infty} \frac{g(n)}{f(n)} = \lim_{n \to \infty} \min\{i \in \mathbb{N} \mid \tilde{n}_i \leq n\} = \infty \, . \tag{44}$$

However, by using equations 40 and 43, we also have:

$$\forall n_0 \in \mathbb{N}, \, \exists n_0 \leq n \in \mathbb{N} : P\left(\|M_n\| \leq g(n)\right) < p \, , \tag{45}$$

which means that:

$$\lim_{n \to \infty} P\left(\|M_n\| \leq g(n)\right) \neq 1 \, . \tag{46}$$

This contradicts our assumption in equation 32. Therefore, by reductio ad impossibile, equation 33 must be satisfied, completing the proof for the second direction. $\square$

### *Proof - Lemma A.5.*

For a trivial lower bound, we choose $h_-$ such that $\forall n \in \mathbb{N} : h_-(n) = 0$.

We define $h_+$ as follows:

$$\forall n \in \mathbb{N} : h_+(n) = \inf\left\{m \in \mathbb{R} \, \middle| \, 1 - \frac{1}{n} \leq P\left(\|M_n\| \leq m\right)\right\} \, . \tag{47}$$

The infimum and the function are well defined because:

1. The set is well defined.

2. The set is non-empty; if it were empty, it would imply that there is some probability that $\|M\|$, which is a positive number, is larger than any real number, which is impossible.

3. The set is defined with a total order " $<$ " and has a lower bound, $m = 0$.

Since for any $0 < p < 1$, there exists some $n_0 \in \mathbb{N}$ such that:

$$\forall n_0 \leq n \in \mathbb{N} : p \leq P\left(\|M_n\| \leq m\right), \tag{48}$$

we know that for any $h_+ < g \in \mathcal{N}$, this is also true, which implies:

$$O(M) \leq O\left(h_+\right), \tag{49}$$

completing the proof. $\qquad\qquad\square$

***Proof - Theorem 2.1.***

General Idea of the Proof:

The proof proceeds as follows:

- We consider the set of all upper bounds for $M$, denoted by $\mathcal{Z}$, and use Zorn's lemma to show that every chain[4] in this set has a lower bound within $\mathcal{Z}$.

- Applying Zorn's lemma again, we demonstrate that $\mathcal{Z}$ has a minimum.

- We then show that the limiting behavior of this minimum is unique.

Existence of an Infimum for the Upper Bound Set:

We begin by defining the set:

$$\mathcal{Z} = \{h \in \mathcal{N} \,|\, O\left(M\right) \leq O\left(h\right)\} \ . \tag{50}$$

This set is:

1. Well defined.

2. Non-empty (as proven in lemma A.5).

3. Defined with a partial order $h_1 < h_2 \leftrightarrow O\left(h_1\right) < O\left(h_2\right)$.

According to Zorn's lemma, if all chains in this set have a lower bound in $\mathcal{Z}$, then $\mathcal{Z}$ has at least one minimum.

Given some chain in the set, $\mathcal{C} \subseteq \mathcal{Z}$, we know it is lower bounded by the function $h_-$, which means (by using Zorn's lemma) it has at least one infimum (a lower bound without any smaller lower bounds). We will choose such an infimum and denote it by $I \in \mathcal{N}$.

Proving that the Infimum is in $\mathcal{Z}$:

We assume, by contradiction, that this infimum is not in $\mathcal{Z}$, which means there exists some $g \in \mathcal{N}$ such that $I = o(g)$ and for every $0 < p < 1, n_0 \in \mathbb{N}$, there exists $n_0 \leq n \in \mathbb{N}$ such that:

$$P\left(\|M_n\| \leq g\left(n\right)\right) < p \ . \tag{51}$$

Since $I = o(g)$, we know that for any $c \in \mathbb{R}$ and sufficiently large $n \in \mathbb{N}$:

$$cI\left(n\right) \leq g\left(n\right) \ . \tag{52}$$

Combining these equations, we obtain:

$$\forall 0 < c, \, n_0 \in \mathbb{N} \,\exists n_0 \leq n \in \mathbb{N} : P\left(\|M_n\| \leq cI\left(n\right)\right) < p \ . \tag{53}$$

In particular, if we choose the series $\forall i \in \mathbb{N} : c_i = i^2$, there exist $\tilde{n}_1 < \tilde{n}_2 < \tilde{n}_3... \in \mathbb{N}$ such that:

$$\forall i \in \mathbb{N} : P\left(\|M_{\tilde{n}_i}\| \leq i^2 I\left(\tilde{n}_i\right)\right) < p \ . \tag{54}$$

We can require that $\{\tilde{n}_i\}_{i=1}^{\infty}$ is increasing for the same reason as before, as we know that we can find such arbitrarily large $n$-values for any sufficiently large $n_0$ and for any $c$, so we can, by induction, demand that each $\tilde{n}_i$ is greater than all previous $\tilde{n}_1...\tilde{n}_{i-1}$.

---

[4]A chain, as defined in set theory, is a subset for which the given partial order becomes a total order.

Now, we define the function:

$$J\left(n\right) = \begin{cases} iI\left(n\right) : \exists i \in \mathbb{N} : n = \tilde{n}_i \\ I\left(n\right) : \text{else} \end{cases} . \tag{55}$$

This function is well defined because there is only one $i$ for any $n$ such that $n = \tilde{n}_i$, as it is an increasing series.

Using equations 54,55, we find that the subseries $\{\tilde{n}_i\}_{i=1}^{\infty}$ satisfies:

$$\forall i \in \mathbb{N} : P\left(\left\| M\left(q\right)_{\tilde{n}_i} \right\| \leq iJ\left(\tilde{n}_i\right)\right) < p . \tag{56}$$

Applying lemma A.4 for the equivalency of the asymptotic bound definition, we conclude that above this subseries $J \notin \mathcal{Z}$, which implies that above this subseries $J$ is a lower bound of $\mathcal{Z}$ and consequently, also of $\mathcal{C}$. Moreover, for all other $n$, we have $J = I$, and since $I$ is a lower bound of $\mathcal{C}$, so is $J$. Since every $n \in \mathbb{N}$ belongs to one of these subseries, we conclude that $J$ is a lower bound of $\mathcal{C}$ in general.

Furthermore, for every $i \in \mathbb{N}$ as $1 \leq c_i$, we have:

$$\forall n : I(n) \leq J(n) \rightarrow O(I) \leq O(J) . \tag{57}$$

However, since $\{\tilde{n}_i\}_{i=1}^{\infty}$ is increasing and unbounded, we know that there exists at least one subseries such that:

$$\lim_{\tilde{n}_i \to \infty} \frac{J\left(\tilde{n}_i\right)}{I\left(\tilde{n}_i\right)} = \lim_{i \to \infty} c_i = \infty \rightarrow O\left(J\right) \neq O\left(I\right) . \tag{58}$$

This implies:

$$O\left(I\right) < O\left(J\right) \rightarrow I < J . \tag{59}$$

We have discovered that $J$ is greater than $I$, but smaller than all functions in $\mathcal{C}$, which implies that it is a larger lower bound than the infimum, which is impossible! and implies by "reductio ad impossibile" that every chain in $\mathcal{Z}$, has a lower bound in $\mathcal{Z}$.

Existence and Uniqueness of the Minimum:

Using Zorn's lemma, we now know that $\mathcal{Z}$ has at least one minimum, denoted by $f \in \mathcal{N}$. Our remaining task is to show that all other minima in $\mathcal{Z}$ exhibit the same limiting behavior as $f$, which implies the uniqueness of the minimal limiting behavior.

Let $g \in \mathcal{N}$ be another minimum. We define:

$$\forall n \in \mathbb{N} : h\left(n\right) = \min\left\{f\left(n\right), g\left(n\right)\right\} . \tag{60}$$

We know that $h \leq f, g$ (as all of its elements are smaller or equal to those of $f, g$), and we also know that $h \in \mathcal{Z}$ since $f, g \in \mathcal{Z}$ and for every $0 < p < 1$ we can choose the maximal $n_0$ from $f$ and $g$. Thus, $h \in \mathcal{Z}$, but $h \leq f, g$ as well, where $f, g$ are minima themselves. This implies:

$$O\left(f\right) = O\left(h\right) = O\left(g\right) \rightarrow O\left(f\right) = O\left(g\right) . \tag{61}$$

Therefore, there exists a unique minimal limiting behavior, which implies that the tensor's asymptotic behavior is always well-defined. □

**Remark A.4.** In our proof, we employed Zorn's lemma twice. First, we used it to demonstrate the existence of an infimum for every chain, and then, after showing that these infima belong to $\mathcal{Z}$, we employed it again to establish that $\mathcal{Z}$ has a minimum. At first glance, it may seem perplexing that we needed to rely on Zorn's lemma, an incredibly abstract and powerful tool equivalent to the somewhat controversial axiom of choice, to prove that the tensor's asymptotic behavior, which has a much more grounded and intuitive meaning, is well-defined.

One possible explanation for this discrepancy is that we may not have actually required the full power of the axiom of choice, and our structures could be simple enough that an alternative approach could have been taken to prove our theorem without using Zorn's lemma. We believe, however, that in the most general case, Zorn's lemma was indeed necessary, but it was only relevant for extreme distributions lacking any tangible "physical meaning." For any well-defined set of distributions with a clear underlying meaning, one could potentially find an alternative method for demonstrating the existence of a tight bound without invoking Zorn's lemma.

In any case, as we demonstrated in Lemma A.5, there is no need for any of these high-level tools to prove the existence of an upper bound.

A.3    PROPERTIES OF THE ASYMPTOTIC BEHAVIOR NOTATION

Having established that our notation is meaningful, we now aim to demonstrate its usefulness. First, we need to address our earlier issue and define "uniform asymptotic bound." Once again, we omit the proofs in this (and next) sections.

**Definition A.1** (Uniform Tensors Asymptotic Bound). Given a series of random tensors $\left\{M^{(d)}\right\}_{d=1}^{D}$, where $D \in \mathbb{N} \cup \{\infty\}$ (or, more precisely, a series of random tensor series) with a limiting parameter $n$, we say that it is uniformly asymptotically upper bounded by $f \in \mathbb{N}$ under some rising monotonic function $\mathcal{K}^{1...D} : \mathbb{R} \to \mathbb{R}$:

$$\forall d = 1...D : O\left(M^{(d)}\right) \leq O\left(\mathcal{K}^d \circ f\right) \quad \text{Uniformly} , \tag{62}$$

if and only if:

$$\forall g \in \mathcal{N} \, s.t \, f = o\,(g) : \lim_{n \to \infty} P\left(\forall d = 1...D : \left\|M_n^{(d)}\right\| \leq \mathcal{K}^d \circ g\,(n)\right) = 1 . \tag{63}$$

The definition for a uniform lower asymptotic bound is analogous with reversed directions.

**Remark A.5.** As discussed in definition 2.1, it is clear that if $D$ is finite, then a uniform bound is equivalent to a point-wise bound.

**Lemma A.6** (Asymptotic Notation Inherits its Norm Properties). Given a random tensor $M$ and a series of jointly distributed random tensors $\left\{M^{(d)}\right\}_{d=1}^{D}$ (with $M$ as well), where $D \in \mathbb{N} \cup \{\infty\}$, such that they are all uniformly bounded:

$$\forall d = 1...D : O\left(M^{(d)}\right) \leq O\left(\mathcal{K}^d \circ f\right) \quad \text{Uniformly} , \tag{64}$$

then:

1. If some positive linear combination of $M^{(d)}$'s norms satisfies an inequality of the form:

$$\|M\| \leq \sum_{\tilde{d}=1}^{\tilde{D}} \lambda_{\tilde{d}} \prod_{d=D_{\tilde{d}-1}+1}^{D_{\tilde{d}}} \|M_d\| , \tag{65}$$

where all of the coefficients are positive: $\forall d = 1...\tilde{D} : 0 \leq \lambda_{\tilde{d}}$ and we divided $1...D$ into a series of finite intervals: $0 = D_1 < D_2 < ... < D_{\tilde{D}} = D$. Then the asymptotic behavior of all the tensors satisfies the same inequality as well for every $h \sim f$:

$$O\,(M) \leq O\left(\sum_{\tilde{d}=1}^{\tilde{D}} \lambda_{\tilde{d}} \prod_{d=D_{\tilde{d}-1}+1}^{D_{\tilde{d}}} \mathcal{K}^d \circ h\right) , \tag{66}$$

and if the inequality is an equality for the norm, it is also an equality for the "large $O$-s."

2. Our asymptotic notation inherits all of the properties presented in lemma A.1.

**Remark A.6.** The lemma still holds even if the tensor have additional indices, as we will see in section (E.4), provided the number of additional index possibilities remains finite in $n$.

A.4    EXPLORING THE RELATIONSHIP BETWEEN ASYMPTOTIC BEHAVIOR NOTATION AND THE TENSORS' MOMENTS

The final aspect of the asymptotic behavior notation we wish to explore is the relationship between this notation and the moments of our tensors' norm or variables. This relationship is relatively intuitive and straightforward, and will be useful in Section (E). We first need to introduce a simple notation for every tensor $M_{\vec{i}}$ that will assist in examining tensor moments, the norm expectation value, defined as:

$$[M] = \sqrt{\frac{1}{N} \left\langle \|M\|^2 \right\rangle} , \tag{67}$$

**Lemma A.7** (Asymptotic Behavior and Tensor Moments Equivalency). Given a random tensor $M$ and a function $f \in \mathcal{N}$, then:

$$O(M) \leq O(f) \,, \tag{68}$$

if and only if with probability arbitrarily close to 1:

$$[M] = O(f) \,. \tag{69}$$

The lemma is also applicable for the uniform bound in the case of infinite number random tensors.

In (4.1), we highlighted that most assertions concerning the convergence of $\mathcal{C}'(F - \hat{y})$ relates to its expected value. However, we can now also associate it with its asymptotic behavior throughout the entire training trajectory. This association stems from the understanding that, if our system exhibits a known average decay, the likelihood of significant deviations from this typical variance range must also decrease, and exponentially (at any decaying rate that is slower then our original rate). Given that decaying geometric sums are convergent, we can infer that the overall probability of the system defying our predicted asymptotic behavior is likewise convergent. Given that we can choose the scaling of this probability arbitrarily, we can set conditions such that the cumulative probability of any deviation is arbitrary small. We introduce this notion for the reader's consideration and propose a detailed formulation as a future exercise.

## B  ADDITIONAL DEFINITIONS

### B.1  DERIVATIVES CORRELATIONS ASYMPTOTIC BEHAVIOR

In our main text (3.2.1), we discussed that the definition for the asymptotic behavior of the derivatives correlations is slightly nuanced, due to the many different potential combinations of distinct inputs. Here we define it rigorously.

**Definition B.1** (Derivatives Correlations Asymptotic Behavior). For every $D \in \mathbb{N}^0$, $d \in \mathbb{N}$, and $d_1 \leq d_2 \leq ... \leq d_{\tilde{d}} \in \mathbb{N}$ such that $d_1 + ... + d_{\tilde{d}} = d$:

$$O_{d_1...d_{\tilde{d}}} \left( \mathfrak{C}^{D,d} \right) \equiv O_{x_0, x_1 ... a_{\tilde{d}} \in \mathcal{P}} \left( \mathfrak{C}^{D,d} \left( x_0, x_1^{\times d_1} ... a_{\tilde{d}}^{\times d_{\tilde{d}}} \right) \right) \,. \tag{70}$$

Inputs order doesn't matter as correlations are symmetric concerning their first derivatives. The factor $\frac{d!}{d_1! \cdots d_{\tilde{d}}!}$ accounts for the possible combinations. If $f \in \mathcal{N}$, we say:

$$\mathfrak{C}^{D,d} = O(f) \,, \tag{71}$$

if and only if all combinations are uniformly bound by $f$. In the continuous limit (extended training time), only $d_1 = ... = d_d = 1$ remains relevant.

### B.2  PROPERLY NORMALISED GDML

Our main theorems (3.1, 3.2 ) and corollary (4.0.1) are applicable for systems that are properly scaled in the initial condition where $n \to \infty$, defined as follows.

**Definition B.2** (PGDML). Given a GDML as described in section 3.1, we will say it is properly normalized and denote it as PGDML if and only if:

$$F(\theta_0) = O(n^0) \tag{72}$$

$$\Delta F(\theta_0) = F(\theta(1)) - F(\theta_0) = O(n^0) \tag{73}$$

$$\mathfrak{C}^1 = (N\eta) O(\nabla F(\theta_0))^2 \tag{74}$$

$$\forall d \in \mathbb{N} : O(\nabla^{\times d} F(\theta_0)) \leq O(\nabla F(\theta_0))^d \quad \text{Uniformly.} \tag{75}$$

The first two conditions (72, 73) ensure that our system scale remains finite for the initial condition. Condition 74 stipulates that the asymptotic behavior of the kernel is maximal, given the asymptotic behavior of the first derivative. This condition ensures that our system is genuinely learning and not only memorizing. This is because the kernel for different inputs is responsible for extrapolation, while the kernel with the same input twice responsible for memorization. Condition 75 asserts that none of the higher derivatives dominate the first for $n \to \infty$, a property that most realistic scalable GDMLs satisfy, because if it is not satisfied, gradient decent becomes irrelevant. We show that wide neural networks in general satisfy that property in appendix E.5.

# C PROOF OF THEOREMS (3.1,3.2)

We can now proceed with the proofs of Theorems 3.1 and 3.2. The general idea has been outlined at the end of section 3.3.1.

## C.1 FIRST DIRECTION OF THEOREMS (3.1,3.2)

Now that we understand how to work with the asymptotic behavior of random tensors, we can proceed to prove our main theorems and corollary. We will begin with the first direction of the theorems.

**Lemma C.1** (Linearization Requires Weak Correlation).

1. In Theorem (3.1), if condition (1) is satisfied, then condition (2) is satisfied as well.

2. In Theorem (3.2), if condition (1) is satisfied, then condition (2) is satisfied as well.

*Proof.* We only demonstrate that the $O_1(\mathfrak{C})$ are bounded; The proof the rest are bounded is the same, by considering more learning steps after the initial condition.

For the initial condition, we know that any reparameterization $0 < r$ satisfies (11,20):

$$
\begin{aligned}
F\left(\theta\left(1\right)\right) - F_{lin}\left(1\right) = \\
\sum_{d=1}^{\infty} \frac{(r\eta)^d}{d!} \left( \nabla^{\times d} F\left(\theta_0\right) \left( \nabla F\left(\theta_0\right) \left(x_1\right)^T \right)^{\times d} \right) \left( -\mathcal{C}'\left( F\left(\theta_0\right)\left(x_1\right), \hat{y}\left(x_1\right) \right) \right)^{\times d} - \\
\left( -\left(r\eta\right) \nabla F\left(\theta_0\right) \nabla F\left(\theta_0\right)\left(x_1\right)^T \left( -\mathcal{C}'\left( F\left(\theta_0\right)\left(x_1\right), \hat{y}\left(x_1\right) \right) \right) \right) = \\
\sum_{d=2}^{\infty} r^d \left( \frac{\eta^d}{d!} \nabla^{\times d} F\left(\theta_0\right) \left( \nabla F\left(\theta_0\right)\left(x_1\right)^T \right)^{\times d} \right) \left( -\mathcal{C}'\left( F\left(\theta_0\right)\left(x_1\right), \hat{y}\left(x_1\right) \right) \right)^{\times d} = \\
\sum_{d=2}^{\infty} r^d \left( \mathfrak{C}^d \right)^{\cdot, x_1^{\times d}} \left( -\mathcal{C}'\left( F\left(\theta_0\right)\left(x_1\right), \hat{y}\left(x_1\right) \right) \right)^{\times d} ,
\end{aligned}
\tag{76}
$$

and in the same way for every $D \in \mathbb{N}$:

$$
\begin{aligned}
\frac{(r\eta)^{\frac{D}{2}}}{D!} \nabla^{\times D} F\left(\theta\left(1\right)\right) - \frac{(r\eta)^{\frac{D}{2}}}{D!} \nabla^{\times D} F_{lin}\left(\theta_0\right) = \\
\sum_{d=1}^{\infty} r^{\frac{D}{2}+d} \left( \mathfrak{C}^{D,d} \right)^{\cdot, x_1^{\times d}} \left( -\mathcal{C}'\left( F\left(\theta_0\right)\left(x_1\right), \hat{y}\left(x_1\right) \right) \right)^{\times d} .
\end{aligned}
\tag{77}
$$

Utilizing Lemma (A.6), it becomes evident that for properly normalized gradient descent-based systems:

$$
O\left( \mathfrak{C}^{D,d} \mathcal{C}'\left( F\left(\theta_0\right), \hat{y} \right)^{\times d} \right) \leq O\left( \mathfrak{C}^{D,d} \right) O\left( \mathcal{C}'\left( F\left(\theta_0\right), \hat{y} \right)^{\times d} \right) = O\left( \mathfrak{C}^{D,d} \right) .
\tag{78}
$$

However, since our theorem should work for any $\hat{y}$, we can choose $U = F\left(\theta_0\right) + c$, and obtain:

$$
O\left( \mathfrak{C}^{D,d} \mathcal{C}'\left( F\left(\theta_0\right), \hat{y} \right)^{\times d} \right) \propto O\left( \mathfrak{C}^{D,d} \mathcal{C}'\left( c \right)^{\times d} \right) = O\left( \mathfrak{C}^{D,d} \right) ,
\tag{79}
$$

as we can choose $c$ such that $\mathcal{C}'(c)$ is the vector that maximizes the correlation, as $\mathcal{C}'$ is convex and the correlations are symmetrical.

Given that we can choose an open set of different scalings of $r$, we know the different elements in the series cannot cancel each other out. Consequently, for $F - F_{lin}$ to decay, all the distinct elements must decay.

Assuming condition (1) in Theorem (3.1):

Given that $O\left( F\left(\theta\left(1\right)\right) - F_{lin}\left(1\right) \right) = O\left( \frac{1}{m(n)} \right)$ and for every $D \in \mathbb{N}$ we have $O\left( \eta^{\frac{D}{2}} \nabla^{\times D} F\left(\theta\left(1\right)\right) - \eta^{\frac{D}{2}} \nabla^{\times D} F\left(\theta_0\right) \right) = O\left( \frac{1}{\sqrt{m(n)}} \right)$, it follows that each correlation must decay at least like:

$$
\forall 2 \leq d \in \mathbb{N} : O\left( \mathfrak{C}^d \right) \leq O\left( \frac{1}{m(n)} \right) \quad \text{Uniformly,}
\tag{80}
$$

and

$$\forall D, d \in \mathbb{N} : O\left(\mathfrak{C}^{D,d}\right) \leq O\left(\frac{1}{\sqrt{m(n)}}\right) \quad \text{Uniformly.} \tag{81}$$

This completes the first part of the proof.

Assuming condition (1) in Theorem (3.2):

By taking $r(n)$ arbitrarily close to $m(n)$, we find that for $F(\theta(1)) - F_{lin}(1)$ to decay, $r^d \mathfrak{C}^d$ must decay as well, which implies that:

$$\forall d \in \mathbb{N} : O\left(\mathfrak{C}^d\right) \leq O\left(\frac{1}{m(n)}\right)^d , \tag{82}$$

and

$$\forall D \in \mathbb{N}^0, d \in \mathbb{N} : O\left(\mathfrak{C}^{D,d}\right) \leq O\left(\frac{1}{\sqrt{m(n)}}\right)^d . \tag{83}$$

This concludes our proof.

$\square$

## C.2 Second Direction of Theorems (3.1,3.2)

We will now prove the other direction of the theorems, focusing on Theorem (3.1) since the proofs for the other theorems are essentially the same. It should also be noted that the corollary (4.0.1), which will be proven next, is almost a generalization of this direction, except that it is only applicable for sufficiently small learning rates.

**Lemma C.2** (Asymptotic Behavior Normalization for weakly Correlated PGDML). Consider a weakly correlated PGDML as described in Theorems (3.1,3.2) then we have:

$$\forall D \in \mathbb{N} : \eta^D O\left(\nabla^{\times D} F(\theta_0)\right)^2 \leq O(1) \quad \text{Uniformly.} \tag{84}$$

With Lemma C.2 at hand, we can now demonstrate the second direction of the theorem by proving a slightly stronger version of it.

**Lemma C.3** (Weak Correlations Create Linearization - First Theorem). Assuming the conditions of Theorem (3.1) part (1), then for every $s = 1...S$:

1.
$$O\left(F(\theta(s)) - F_{lin}(s)\right) \leq O\left(\frac{1}{m(n)}\right) . \tag{85}$$

2.
$$O\left(\eta^{\frac{1}{2}} \nabla F(\theta(s)) - \eta^{\frac{1}{2}} \nabla F(\theta_0)\right) \leq \gamma . \tag{86}$$

3. For every $2 \leq D \in \mathbb{N}$

$$O\left(\eta^{\frac{D}{2}} \nabla^{\times D} F(\theta(s)) - \eta^{\frac{D}{2}} \nabla^{\times D} F(\theta_0)\right) \leq O\left(\frac{1}{\sqrt{m(n)}}\right) \quad \text{uniformly.} \tag{87}$$

Here, $\gamma$ is an asymptotic notation such that $\gamma = O\left(\frac{1}{\sqrt{m(n)}}\right)$, and when multiplied with a first derivative of the hypothesis function in its initial condition, it exhibits an asymptotic behavior of $O\left(\gamma_t \eta^{\frac{1}{2}} \nabla F(\theta_0)\right) \leq O\left(\frac{1}{m(n)}\right)$.

From proving lemmas (C.1,C.3), we can conclude that theorems (3.1,3.2) have been proven.

***Proof of Lemma (C.2).***

Assume that the lemma is not satisfied, i.e.,

$$\eta O \left(\nabla F \left(\theta_0\right)\right)^2 \not\lesssim O \left(1\right) , \tag{88}$$

then for some probability $0 < p < 1$, we have:

$$O \left(1\right) < \eta O \left(\nabla F \left(\theta_0\right)\right)^2 . \tag{89}$$

Utilizing the third property of PGDML systems (74), we conclude that for some relevant probability:

$$O \left(1\right) < O \left(\mathfrak{C}^1\right) . \tag{90}$$

However, for the reasons discussed earlier, the different elements in the equation of motion cannot cancel each other out, as $\eta$ can be chosen from an open set. This implies that the second property of PGDML systems (73) cannot be satisfied, leading to the conclusion that:

$$\eta O \left(\nabla F \left(\theta_0\right)\right)^2 \leq O \left(1\right) , \tag{91}$$

must hold.

By employing the fourth property (75) of PGDML systems, we obtain the desired result.

$\square$

***Proof of Lemma (C.3).***

We will prove the lemma using induction over the learning steps (of course). The induction base for the "zero" step, where $\theta = \theta_0$, is trivial. Assuming the lemma holds for $s \in \mathbb{N}^0$, we observe that for every $\left(D \in \mathbb{N}^0, d \in \mathbb{N}\right) \neq \left(0, 1\right)$, the $d, D$ correlation satisfies the following for sufficiently small learning rate $\eta$:

$$\mathfrak{C}^{D,d} \left(\theta \left(s\right)\right) = \eta^{\frac{D}{2}+d} \nabla^{\times D+d} F \left(\theta \left(s\right)\right)^T \nabla F \left(\theta \left(s\right)\right)^{\times d}$$
$$=$$
$$\left(\eta^{\frac{D+d}{2}} \nabla^{\times D+d} F \left(\theta_0\right) + \gamma\right)^T \left(\eta^{\frac{1}{2}} \nabla F \left(\theta_0\right) + \gamma\right)^{\times d}$$
$$=$$
$$\mathfrak{C}^{D,d} + \gamma^T \left(\eta^{\frac{1}{2}} \nabla F \left(\theta_0\right)\right)^{\times d} + \gamma^T \left(\gamma \times \left(\eta^{\frac{1}{2}} \nabla F \left(\theta_0\right)\right)^{\times d-1}\right) +$$
$$\eta^{\frac{D+d}{2}} \nabla^{\times D+d} F \left(\theta_0\right) \left(\gamma \times \left(\eta^{\frac{1}{2}} \nabla F \left(\theta_0\right)\right)^{\times d-1}\right) + \text{comb} + O \left(\frac{1}{m(n)}\right)$$
$$=$$
$$\mathfrak{C}^{D,d} + O \left(\frac{1}{m(n)}\right) + O \left(\frac{1}{m(n)}\right) + d \mathfrak{C}^{D+1,d-1} \times \gamma + O \left(\frac{1}{m(n)}\right)$$
$$=$$
$$\mathfrak{C}^{D,d} + O \left(\frac{1}{m(n)}\right) . \tag{92}$$

Here, we used the derivatives correlation definition, lemmas, the induction hypothesis, the bound of the correlations from condition (1), and the definition of $\gamma$.

By employing the derivative's correlation definition and condition (1), we observe that:

$$\forall 2 \leq d \in \mathbb{N} : O \left(\mathfrak{C}^d\right) = O \left(\frac{1}{m(n)}\right) ,$$
$$\forall d \in \mathbb{N} : O \left(\mathfrak{C}^{1,d}\right) = \gamma ,$$
$$\forall 2 \leq D \in \mathbb{N}, d \in \mathbb{N} : O \left(\mathfrak{C}^{D,d}\right) = O \left(\frac{1}{\sqrt{m(n)}}\right) . \tag{93}$$

Furthermore:

$$\mathfrak{C}^{1,d} \eta^{\frac{1}{2}} \nabla F \left(\theta_0\right) = \eta^{\frac{1}{2}+d} \nabla^{\times d+1} F \left(\theta_0\right)^T \left(\eta^{\frac{1}{2}} \nabla F \left(\theta_0\right)\right)^{\times d} =$$
$$\eta^{d+1} \nabla^{\times d+1} F \left(\theta_0\right)^T \left(\nabla F \left(\theta_0\right)\right)^{\times d+1} = \mathfrak{C}^{d+1} . \tag{94}$$

Hence, using this equation, we can deduce that $\mathfrak{C}^{D,d} \left(\theta \left(s+1\right)\right)$ satisfies the given conditions as well. By incorporating this equation into our equation of motion and employing the lemmas, we find that for a sufficiently small learning rate, $F \left(\theta \left(s+1\right)\right)$ also satisfies the lemma. Consequently, by induction, the lemma holds for all $s \in \mathbb{N}$. $\square$

# D   PROOF OF COROLLARY 4.0.1

In this section, we prove corollary 4.0.1. The general approach for this proof is akin to that of the first direction of Theorems 3.1 and 3.2, albeit with an additional focus on the evolution of the deviation throughout the induction process.

Given the complexity of tracking all the derivatives simultaneously, our strategy involves monitoring the difference between the parameters and their linearization, as expressed in Equation (101). A significant challenge arises in solving the equation of motion that these parameters must satisfy.

To circumvent this issue, we establish a link between this deviation and the deviation of the generalization function from its linearization (101) up to the highest order, as outlined in equation 106. By considering only the lowest order terms, we obtain an equation of motion (113). In cases where the cost function decays exponentially, and we are able to bound the deviation of this equation.

## D.1   RELATIONS BETWEEN DIFFERENT LINEARIZATIONS

In the main text, we linearised $F$ as $F_{lin}$ (11), by first considering only the linear part of $F$, and then examining how it changes over time for a given training path. However, there are alternative ways to linearise $F$ that can be useful to consider. One such method involves taking only the linear part of $F$, without considering the training path:

$$\hat{F}(\theta) = F(\theta_0) + \nabla F(\theta_0)^T (\theta - \theta_0) . \tag{95}$$

Another useful definition is to examine how $\theta$ would develop over time under the linear approximation for our training path:

$$\begin{aligned}\theta_{lin}(0) = \theta_0 \quad \forall s \in \mathbb{N} : \\ \theta_{lin}(s+1) = \theta_{lin}(s) - \nabla F(\theta_0)(x_s) \mathcal{C}'(F_{lin}(s)(x_s) - \hat{y}(x_s)) .\end{aligned} \tag{96}$$

It can be observed that $F_{lin}, \hat{F}, \theta_{lin}$ satisfy the following relation:

$$\forall s \in \mathbb{N}^0 : F_{lin}(s) = \hat{F}(\theta_{lin}(s)) . \tag{97}$$

A more refined relation is the one between $F(\theta_{lin})$ and $F_{lin}(\theta)$, defined for every $s = 0...S$ as follows:

$$O(F(\theta_{lin}(s)) - F_{lin}(s)) \leq O\left(\frac{\varrho^2(s)}{m(n)}\right) , \tag{98}$$

where $\varrho$ is defined as:

**Definition D.1** (Typical Linear Cumulative Deviation). We define the typical linear cumulative deviation as the bound of the cumulative deviation of $F_{lin}$ from $\hat{y}$:

$$O(\varrho(s)) = \sum_{s'=0}^{s-1} O(\mathcal{C}'(F_{lin}(s_1) - \hat{y})) , \tag{99}$$

and in our case:

$$O(\varrho(s)) \leq O\left(\frac{1 - e^{-\frac{s}{T}}}{1 - e^{-\frac{1}{T}}}\right) \leq O(1) . \tag{100}$$

This implies that $\varrho(s) = o(m(n))$, which is essential for proving (98). We will not provide this proof here, as we will not use it directly in the remainder of this paper, and we will soon prove many similar identities.

## D.2   SMALL PERTURBATION FROM THE LINEAR SOLUTION

The initial approach of the proof aimed to demonstrate that $F$ only deviates slightly from $F_{lin}$, and that also its derivatives deviate slightly at the initial conditions. The intention was to use induction to show that this holds at each time step. This method is effective if the goal is merely to prove that $F$ converges to $F_{lin}$ at a rate of $O\left(\frac{1}{m(n)}\right)$ for a fixed time step. However, it poses challenges when attempting to understand how the two functions deviate from each other over time. This is due to the

necessity of simultaneously tracking the evolution of all derivatives and the changes in correlations over time, which is nearly impossible.

To circumvent this issue, rather than tracking all derivatives, we will calculate how $F(\theta(s))$ deviates from $F_{lin}(s)$ by utilizing a similar relationship to the one we discovered between $\theta_{lin}$ and $F_{lin}$. This will allow us to establish bounds on $F - F_{lin}$. Although the two approaches are equivalent, and the first one is more intuitively clear, the second approach simplifies accurate and simple calculations by focusing on a single object, $F - F_{lin}$.

In the following lemma, we demonstrate how a small perturbation at a given step ($s = 0...S - 1$) results in a small perturbation at the subsequent step ($s + 1$). Then, we will use these results to inductively show the deviation in time between the hypothesis function and its linear approximation.

We denote:
$$\delta(s) = F(\theta(s)) - F_{lin}(s) \,, \; \eta^{\frac{1}{2}}\zeta(s) = \theta(s) - \theta_{lin}(s) \,, \tag{101}$$
and assume that the deviation from linearity is small, hence:
$$O(\delta(s)) \leq O\left(\frac{f(s)}{m(n)}\right) \,, \; O(\zeta(s)) \leq O(g(s))\gamma \,, \tag{102}$$

where
$$f(s), g(s)^2, \varrho(s)^2 = o(m(n)) \,. \tag{103}$$

For some parts of our lemma, it will also be relevant to separate the deviation of the parameters into two components:
$$\zeta(s) = \zeta_\gamma(s) + \zeta_m(s) \,, \tag{104}$$
such that:
$$O(\zeta_\gamma(s)) \leq O(g_\gamma(s))\gamma \,, \; O(\zeta_m(s)) \leq O\left(\frac{g_m(s)}{m(n)}\right) \,. \tag{105}$$

**Remark D.1.** Here, we consider the case of a general rate of convergence for $\mathcal{C}'(F_{lin}, \hat{y})$, rather than exclusively focusing on an exponential one. This is done to simplify the generalization of our results for reader.

**Lemma D.1** (Deviation of the parameters and of the hypothesis function relations)**.** Given the conditions described above, then up to the leading order:

1.
$$\delta(s) = F(\theta(s)) - F_{lin}(s) \simeq \eta^{\frac{1}{2}}\nabla F(\theta_0)^T \zeta_m(s) + \eta^{\frac{1}{2}}\nabla F(\theta_0)^T \zeta_\gamma(s) +$$
$$\sum_{s_1,s_2=0}^{s-1} \mathfrak{C}^2 \mathcal{C}'(F_{lin}(s_1),\hat{y}) \times \mathcal{C}'(F_{lin}(s_2),\hat{y}) + \tag{106}$$
$$2\sum_{s'=0}^{s-1} \mathfrak{C}^{1,1}\zeta_\gamma(s)\mathcal{C}'(F_{lin}(s'),\hat{y}) + \eta\nabla^{\times 2}F(\theta_0)^T \zeta_\gamma(s)^{\times 2} \,,$$
which means:
$$O(\delta(s)) = O(F(\theta(s)) - F_{lin}(s)) \leq$$
$$O\left(\frac{g_m(s)}{m(n)}\right) + O\left(\frac{(g_\gamma(s)+\varrho(s))^2}{m(n)}\right) \leq O\left(\frac{(g(s)+\varrho(s))^2}{m(n)}\right) \,. \tag{107}$$

2.
$$O\left(\eta^{\frac{1}{2}}\nabla F(\theta_0)^T - \eta^{\frac{1}{2}}\nabla F(\theta_0)^T\right) \leq O(g(s)+\varrho(s))\gamma \,. \tag{108}$$

3.
$$\mathcal{C}'(F(\theta(s)),\hat{y}) - \mathcal{C}'(F_{lin}(s),\hat{y}) \simeq \mathcal{C}''(F_{lin}(s),\hat{y})\delta(s) \,. \tag{109}$$
where $\mathcal{C}''(F_{lin}(s),\hat{y})$ is a positive random matrix such as if the asymptotic behavior of $\mathcal{C}'(F_{lin}(s),\hat{y})$ is bound (as in our case), so is $\mathcal{C}''(F_{lin}(s),\hat{y})$.

4.
$$\eta^{\frac{1}{2}}\zeta(s+1) - \eta^{\frac{1}{2}}\zeta(s) = \theta(s+1) - \theta_{lin}(s+1) - \eta^{\frac{1}{2}}\zeta(s) \simeq$$
$$-\eta\nabla F(\theta_0)\mathcal{C}''(F_{lin}(s'),\hat{y})\delta(s) + O(g(s)+\varrho(s))\mathcal{C}'(F_{lin}(s),\hat{y})\eta^{\frac{1}{2}}\gamma \,, \tag{110}$$
which means:
$$O(\zeta(s+1) - \zeta(s)) \leq O\left(\frac{f(s)}{m(n)}\right) + O(\mathcal{C}'(F_{lin}(s),\hat{y}))O(g(s)+\varrho(s))\gamma \,. \tag{111}$$

5.

$$O\left(\delta\left(s+1\right)-\delta\left(s\right)+\Theta_0\mathcal{C}''\left(F_{lin}\left(s\right),\hat{y}\right)\delta\left(s\right)\right)\leq$$
$$O\left(\frac{\left(g(s)+\varrho(s)\right)^2}{m(n)}\right)O\left(\mathcal{C}'\left(F_{lin}\left(s\right),\hat{y}\right)\right).\tag{112}$$

**Remark D.2.** An important note for our proofs is that all of these components can be generalized to the case where $\zeta(s),\delta(s)$ are not the "original" deviations, as long as they satisfy equation (102).

We can now use this result to prove corollary (4.0.1) by induction. In fact, for the conditions of the corollary at $s = 0$, the induction hypothesis is trivially satisfied as $F(\theta)(0) = F_{lin}(0), \theta(0) = \theta_{lin}(0)$. It is straightforward to show that the contributions of the part multiplied by $O\left(\mathcal{C}'\left(F_{lin}\left(s\right),\hat{y}\right)\right)$ are irrelevant for the possible deviation, as $\mathcal{C}'\left(F_{lin}\left(s\right),\hat{y}\right)\to 0$, $O\left(\varrho\left(s\right)\right)\leq O\left(1\right)$. Consequently, we are left with equations of motion for the asymptotic behavior of the form:

$$O\left(\zeta\left(s+1\right)-\zeta\left(s\right)\right)\leq O\left(\frac{f\left(s\right)}{m\left(n\right)}\right)\quad\delta\left(s+1\right)-\delta\left(s\right)+\Theta_0\mathcal{C}''\left(F_{lin}\left(s\right),\hat{y}\right)\delta\left(s\right)\simeq 0.\tag{113}$$

However, $\Theta_0,\mathcal{C}''$ are positively defined bound matrices, so for a learning rate that is sufficiently small (which would be of the same order of magnitude as the learning rate needed for our system to consistently learning, and for the case where $\mathcal{C}(x) = \frac{1}{2}x^2$, exactly the same), we find that on average this term can only contribute to the shrinkage of $\delta(s)$. This means that neglecting this term for large $s$ would provide an upper bound for the rate of deviation. Thus, we have discovered that the asymptotic behavior of $\delta$ (and consequently, $\zeta$) with respect to time is for large $s$ is bounded by:

$$\delta\left(s+1\right)-\delta\left(s\right)\simeq 0.\tag{114}$$

**This proves our corollary.**

*Proof.*

Part - (1):

$$F\left(\theta\left(s\right)\right) = F\left(\theta_{lin}\left(s\right)+\eta^{\frac{1}{2}}\zeta\left(s\right)\right) =_1$$
$$F\left(\theta_0 - \eta\sum_{s'=0}^{s-1}\nabla F\left(\theta_0\right)\mathcal{C}'\left(F_{lin}\left(s'\right),\hat{y}\right)+\eta^{\frac{1}{2}}\zeta\left(s\right)\right) =_2$$
$$F\left(\theta_0\right) - \sum_{s'=0}^{s-1}\mathfrak{C}^1\mathcal{C}'\left(F_{lin}\left(s'\right),\hat{y}\right)+\eta^{\frac{1}{2}}\nabla F\left(\theta_0\right)^T\zeta\left(s\right)+$$
$$\sum_{s_1,s_2=0}^{s-1}\mathfrak{C}^2\mathcal{C}'\left(F_{lin}\left(s_1\right),\hat{y}\right)\times\mathcal{C}'\left(F_{lin}\left(s_2\right),\hat{y}\right)+$$
$$2\sum_{s'=0}^{s-1}\mathfrak{C}^{1,1}\zeta\left(s\right)\mathcal{C}'\left(F_{lin}\left(s'\right),\hat{y}\right)+\eta\nabla^{\times 2}F\left(\theta_0\right)^T\zeta\left(s\right)^{\times 2}+\ldots\simeq_3$$
$$F_{lin}\left(s\right)+\eta^{\frac{1}{2}}\nabla F\left(\theta_0\right)^T\zeta\left(s\right)+\sum_{s_1,s_2=0}^{s-1}\mathfrak{C}^2\mathcal{C}'\left(F_{lin}\left(s_1\right),\hat{y}\right)\times\mathcal{C}'\left(F_{lin}\left(s_2\right),\hat{y}\right)+$$
$$2\sum_{s'=0}^{s-1}\mathfrak{C}^{1,1}\zeta\left(s\right)\mathcal{C}'\left(F_{lin}\left(s'\right),\hat{y}\right)+\eta\nabla^{\times 2}F\left(\theta_0\right)^T\zeta\left(s\right)^{\times 2},\tag{115}$$

where in (1) we used equation the definition of $\theta_{lin}$ (96), in (2) we expanded our generalisation function as a Taylor sires, and the definition of the derivatives correlations (3.1). In (3) we used the fact that under our assumptions our system is exponentially weakly correlated. Using this result we get our desired identity.

Subtracting $F_{lin}$ we get using the weak derivatives correlations property that up to the leading order:

$$F\left(\theta\left(s\right)\right) - F_{lin}\left(s\right) \simeq$$
$$\eta^{\frac{1}{2}}\nabla F\left(\theta_0\right)^T \zeta\left(s\right) + \sum_{s_1,s_2=0}^{s-1} \mathfrak{C}^2 \mathcal{C}'\left(F_{lin}\left(s_1\right),\hat{y}\right) \times \mathcal{C}'\left(F_{lin}\left(s_2\right),\hat{y}\right) +$$
$$2\sum_{s'=0}^{s-1} \mathfrak{C}^{1,1}\zeta\left(s\right)\mathcal{C}'\left(F_{lin}\left(s'\right),\hat{y}\right) + \eta\nabla^{\times 2}F\left(\theta_0\right)^T \zeta\left(s\right)^{\times 2} \simeq$$
$$\eta^{\frac{1}{2}}\nabla F\left(\theta_0\right)^T \zeta_m\left(s\right) + \eta^{\frac{1}{2}}\nabla F\left(\theta_0\right)^T \zeta_\gamma\left(s\right) +$$
$$\sum_{s_1,s_2=0}^{s-1} \mathfrak{C}^2 \mathcal{C}'\left(F_{lin}\left(s_1\right),\hat{y}\right) \times \mathcal{C}'\left(F_{lin}\left(s_2\right),\hat{y}\right) + \tag{116}$$
$$2\sum_{s'=0}^{s-1} \mathfrak{C}^{1,1}\zeta_\gamma\left(s\right)\mathcal{C}'\left(F_{lin}\left(s'\right),\hat{y}\right) + \eta\nabla^{\times 2}F\left(\theta_0\right)^T \zeta_\gamma\left(s\right)^{\times 2} =$$
$$O\left(\frac{g_m(s)+g_\gamma(s)}{m(n)}\right) + O\left(\frac{\varrho(s)^2}{m(n)}\right) + 2O\left(\frac{\varrho(s)g_\gamma(s)}{m(n)}\right) + O\left(\frac{g_\gamma^2(s)}{m(n)}\right) =$$
$$O\left(\frac{g_m(s)}{m(n)}\right) + O\left(\frac{(g_\gamma(s)+\varrho(s))^2}{m(n)}\right) \leq O\left(\frac{(g(s)+\varrho(s))^2}{m(n)}\right),$$

which finishes our proof.

Part - (2):

Using the same ideas we get:

$$\eta^{\frac{1}{2}}\nabla F\left(\theta_0\right)^T = \eta^{\frac{1}{2}}\nabla F\left(\theta_{lin}\left(s\right) + \eta^{\frac{1}{2}}\zeta\left(s\right)\right)^T =$$
$$\eta^{\frac{1}{2}}\nabla_T F\left(\theta_0 - \eta\sum_{s'=0}^{s-1}\nabla F\left(\theta_0\right)\mathcal{C}'\left(F_{lin}\left(s'\right),\hat{y}\right) + \eta^{\frac{1}{2}}\zeta\left(s\right)\right) = \tag{117}$$
$$\eta^{\frac{1}{2}}\nabla F\left(\theta_0\right)^T - \sum_{s'=0}^{s-1}\mathfrak{C}^{1,1}\mathcal{C}'\left(F_{lin}\left(s'\right),\hat{y}\right) + \eta\nabla^{\times 2}F\left(\theta_0\right)^T\zeta\left(s\right) + \ldots =$$
$$\eta^{\frac{1}{2}}\nabla F\left(\theta_0\right)^T + O\left(\varrho\left(s\right)\right)\gamma_t + O\left(g\left(s\right)\right)\gamma_t.$$

Taking transpose on both sides we get finish our proof.

Part (3):

Using the definition of $\delta$ and the fact that $\mathcal{C}$ is analytical we know that up to the highest order:

$$\mathcal{C}'\left(F\left(\theta\left(s\right)\right),\hat{y}\right) = \mathcal{C}'\left(F_{lin}\left(s\right) + \delta\left(s\right),\hat{y}\right) \simeq \mathcal{C}'\left(F_{lin}\left(s\right),\hat{y}\right) + \mathcal{C}''\left(F_{lin}\left(s\right),\hat{y}\right)\delta\left(s\right) \tag{118}$$

and as $\mathcal{C}$ is convex (), we know that it's second derivative is always a positive matrix. And by using lemma () we know that if the first derivative is bound, so is the second derivative.

Part (4):

Using the equation of motion for $\theta$ (96), and parts (2,3) of this lemma we get that up to leading order:

$$\theta\left(s+1\right) = \theta\left(s\right) - \eta\nabla F\left(\theta\left(s\right)\right)\mathcal{C}'\left(F\left(\theta\left(s\right)\right),\hat{y}\right) \simeq$$
$$\theta\left(s\right) - \eta\left(\begin{array}{c}\nabla F\left(\theta_0\right) + \\ O\left(g\left(s\right)+\varrho\left(s\right)\right)\eta^{\frac{1}{2}}\gamma\end{array}\right)\left(\begin{array}{c}\mathcal{C}'\left(F_{lin}\left(s\right),\hat{y}\right) + \\ \mathcal{C}''\left(F_{lin}\left(s'\right),\hat{y}\right)\delta\left(s\right)\end{array}\right) \simeq \tag{119}$$
$$\theta\left(s\right) - \eta\nabla F\left(\theta_0\right)\mathcal{C}'\left(F_{lin}\left(s\right),\hat{y}\right) - \eta\nabla F\left(\theta_0\right)\mathcal{C}''\left(F_{lin}\left(s'\right),\hat{y}\right)\delta\left(s\right) +$$
$$O\left(g\left(s\right)+\varrho\left(s\right)\right)\mathcal{C}'\left(F_{lin}\left(s\right),\hat{y}\right)\eta^{\frac{1}{2}}\gamma$$

and as:

$$\theta\left(s\right) - \eta\nabla F\left(\theta\left(s\right)\right)\mathcal{C}'\left(F\left(\theta\left(s\right)\right),\hat{y}\right) =$$
$$\theta_{lin}\left(s\right) + \eta^{\frac{1}{2}}\zeta\left(s\right) - \eta\nabla F\left(\theta\left(s\right)\right)\mathcal{C}'\left(F\left(\theta\left(s\right)\right),\hat{y}\right) = \theta_{lin}\left(s+1\right) + \eta^{\frac{1}{2}}\zeta\left(s\right), \tag{120}$$

we get the desired result.

Part (5):

Using the equation of motion for $\theta$, one can see that:

$$
\begin{aligned}
F\left(\theta\left(s+1\right)\right) = F\left(\begin{array}{c} \theta_{lin}\left(s+1\right) - \eta\nabla F\left(\theta_0\right)\mathcal{C}''\left(F_{lin}\left(s'\right),\hat{y}\right)\delta\left(s\right) + \\ \eta^{\frac{1}{2}}\zeta\left(s\right) + O\left(g\left(s\right)+\varrho\left(s\right)\right)\mathcal{C}'\left(F_{lin}\left(s\right),\hat{y}\right)\eta^{\frac{1}{2}}\gamma \end{array}\right) \\
\overset{\simeq_1}{} \\
F_{lin}\left(s+1\right) - \eta\nabla F\left(\theta_0\right)^T\nabla F\left(\theta_0\right)\mathcal{C}''\left(F_{lin}\left(s'\right),\hat{y}\right)\delta\left(s\right) + \\
\eta^{\frac{1}{2}}\nabla F\left(\theta_0\right)^T\zeta\left(s\right) + O\left(g\left(s\right)+\varrho\left(s\right)\right)\mathcal{C}'\left(F_{lin}\left(s\right),\hat{y}\right)\eta^{\frac{1}{2}}\nabla F\left(\theta_0\right)^T\gamma + \\
\sum_{s_1,s_2=0}^{s-1}\mathfrak{C}^2\mathcal{C}'\left(F_{lin}\left(s_1\right),\hat{y}\right)\times\mathcal{C}'\left(F_{lin}\left(s_2\right),\hat{y}\right) + \\
2\sum_{s'=0}^{s-1}\mathfrak{C}^{1,1}\zeta\left(s\right)\mathcal{C}'\left(F_{lin}\left(s'\right),\hat{y}\right) + \\
2O\left(g\left(s\right)+\varrho\left(s\right)\right)\mathcal{C}'\left(F_{lin}\left(s\right),\hat{y}\right)\sum_{s'=0}^{s-1}\mathfrak{C}^{1,1}\gamma\mathcal{C}'\left(F_{lin}\left(s'\right),\hat{y}\right) + \\
\eta\nabla^{\times 2}F\left(\theta_0\right)^T\zeta\left(s\right)^{\times 2} + O\left(g\left(s\right)+\varrho\left(s\right)\right)^2\mathcal{C}'\left(F_{lin}\left(s\right),\hat{y}\right)^2\eta\nabla^{\times 2}F\left(\theta_0\right)^T\gamma^{\times 2} + \\
2O\left(g\left(s\right)+\varrho\left(s\right)\right)\mathcal{C}'\left(F_{lin}\left(s\right),\hat{y}\right)\eta\nabla^{\times 2}F\left(\theta_0\right)^T\left(\gamma\times\zeta\left(s\right)\right) \\
\overset{\simeq_2}{} \\
F_{lin}\left(s+1\right) - \Theta_0\mathcal{C}''\left(F_{lin}\left(s'\right),\hat{y}\right)\delta\left(s\right) + \delta\left(s\right) + \\
2O\left(\frac{g(s)+\varrho(s)}{m(n)}\right)O\left(\mathcal{C}'\left(F_{lin}\left(s\right),\hat{y}\right)\right) + O\left(\frac{(g(s)+\varrho(s))^2}{m(n)}\right)O\left(\mathcal{C}'\left(F_{lin}\left(s\right),\hat{y}\right)\right)^2 + \\
2O\left(\frac{g^2(s)+\varrho(s)g(s)}{m(n)}\right)O\left(\mathcal{C}'\left(F_{lin}\left(s\right),\hat{y}\right)\right) .
\end{aligned}
\tag{121}
$$

where in (1) We use part (1) of the lemma, when we remembered that $O(\delta)\leq O\left(\frac{1}{m(n)}\right)$ so it can be consider as $\zeta_m$. In part (2) we use the definition of $F_{lin}$, $\Theta_0$ and part (1) once again where we gathered all of the components that have only $\zeta(s)$ to get $\delta(s)$. Then we just used the asymptotic behavior of all of the components and took the "worst case scenario" to get equation (112). $\qquad\square$

# E  WIDE NEURAL NETWORKS ARE WEAKLY CORRELATED PGDML SYSTEMS

## E.1  GENERAL IDEA

We start with fully connected neural networks. Although the proof is technically intricate, its underlying concept is straightforward: For the first layer, we observe that all higher correlations exhibit the appropriate asymptotic behavior. We then proceed to prove by induction that all layers manifest the same asymptotic behavior. Consider the second correlation, for instance, which we analyze as follows:

For any general layer $l = 1, \ldots, L$, defining $\nabla_{-l}$ as the derivatives with respect to parameters from layers 1 to $l-1$ (E.2), we employ the equation for fully connected neural networks (127):

$$
\begin{aligned}
l = 0, \ldots, L : F^{(l)} = \theta^{(l,l-1)}\phi\left(F^{(l-1)}\right) + \theta^{(l)} , \\
\forall x \in X : F(\theta)(x) = F^{(L)}(x), \quad F^{(0)}(x) = a ,
\end{aligned}
\tag{122}
$$

to demonstrate that:

$$
\begin{aligned}
\nabla_{(-l)}^{\times 2}F^{(l)} = \nabla_{(-l)}^{\times 2}\left(\theta^{(l,l-1)}\phi\left(F^{(l-1)}\right) + \theta^{(l)}\right) = \\
\nabla_{(-l)}\times\nabla_{(-l)}\left(\theta^{(l,l-1)}\phi\left(F^{(l-1)}\right) + \theta^{(l)}\right) = \nabla_{(-l)}\times\left(\theta^{(l,l-1)}\nabla_{(-l)}\phi\left(F^{(l-1)}\right)\right) = \\
\nabla_{(-l)}\times\left(\theta^{(l,l-1)}\phi'\left(F^{(l-1)}\right)\nabla_{(-l)}F^{(l-1)}\right) = \\
\theta^{(l,l-1)}\phi''\left(F^{(l-1)}\right)\nabla_{(-l)}F^{(l-1)}\times\nabla_{(-l)}F^{(l-1)} + \theta^{(l,l-1)}\phi'\left(F^{(l-1)}\right)\nabla_{(-l)}^{\times 2}F^{(l-1)}
\end{aligned}
\tag{123}
$$

Consequently, the contribution to the $l$-th correlation (13) from this part is proportional to:

$$
\theta^{(l,l-1)}\phi''\left(F^{(l-1)}\right)\mathfrak{C}_{(l-1)}^1\times\mathfrak{C}_{(l-1)}^1 + \theta^{(l,l-1)}\phi'\left(F^{(l-1)}\right)\mathfrak{C}_{(l-1)}^2 .
\tag{124}
$$

Here we have two terms. We can show the right-hand term is small simply by induction. The proof that the left-hand term is also small is more complex, involving the demonstration that for all hidden layers, the relevant contribution from the first correlation originates from its diagonal terms, i.e., $(\mathfrak{C}^1_{(-l)})_{ii}$.

We can now show that in the term, the left index is identical for both correlations, which follows that for most indices, the relevant terms are offset by the irrelevant ones, keeping our expression small.

For the case that one of the derivative does not belong to layers $l = 1$ to $l - 1$, we explicitly show this term to be negligible, as for most indices it simply resets:

$$\nabla_{i^l i^{l-1}} F_i^{(l)} \propto \delta_{i^l i} \tag{125}$$

In the general case of the $D$-th correlation, while there is some complexity in tracing the combinatorial terms from various combinations of derivatives, the fundamental principle remains consistent.

The generalization of this approach for other architectures is discussed in Section E.5.

### E.2 Asymptotic Behavior of Wide FCN at Initialisation

**Remark E.1.** Throughout this paper we considered $\|M\|$ or $O(M)$ as our way to evaluate the size of our random tensors. But here we mainly consider the normalised terms instead:

$$\frac{1}{\sqrt{N}} \|M\| \quad \text{and} \quad \frac{1}{\sqrt{N}} O(M) \ . \tag{126}$$

This is because, in practice, what we are interested of is the average asymptotic behavior of a tensor, and not the accumulative one.

Fully connected neural networks of depth $2 \le L \in \mathbb{N}$, characterized by $L$ parameter vectors (the biases $\theta^{(1)}, \ldots, \theta^{(L)}$), and $L$ parameter matrices (the weights $\theta^{(L,L-1)}, \ldots, \theta^{(1,0)}$), such as:

$$\begin{aligned} l = 0, ..., L : F^{(l)} &= \theta^{(l,l-1)} \phi\left(F^{(l-1)}\right) + \theta^{(l)} \ , \\ \forall x \in X : F(\theta)(x) &= F^{(L)}(x), \quad F^{(0)}(x) = a \ . \end{aligned} \tag{127}$$

In this representation, $F^{(0)}, F^{(1)}, \ldots, F^{(L-1)}$, and $F^{(L)}$ constitute the input, inner, and output layers, respectively. The activation function $\phi$ is analytical, and all of its derivatives are bounded as described in (23).

**Remark E.2.** Generally when working with FNC we do not operate the activation function over the zero layer, the input. But to make the induction slightly easier, we will simplify our expression such as $\phi$ operates over all layers. It makes no real difference

We focus on "wide" neural networks where the depth $L$ is fixed. As long as $L = O(\log(n))$, we can expect an NTK-like behavior for large $n$, but for simplicity, we focus on the scenario where $L$ remains constant in $n$. We introduce a limiting parameter $n \in \mathbb{N}$ such that the width of all the hidden layers satisfies $n \le n_1, \ldots, n_{L-1}$. To simplify our work, we will amend this assumption by postulating that all layers exhibit the same asymptotic behavior of $n$ - $n_1, \ldots, n_{L-1} \sim n$. This modification does not affect our theorems and lemmas, as it merely establishes a lower bound of our original assumption. As the sizes of the zeroth and last layer are constant (the dimensions of the input and output layers stay fixed in $n$ of course), we arrive at:

$$n_1, \ldots, n_{L-1} \sim n \quad \text{and} \quad n_0, n_L \sim 1 \ . \tag{128}$$

Back in the 1960s, it was demonstrated that with Gaussian initialization, we can keep our layers normalised by selecting initial parameters as follows:

$$\forall l = 1, \ldots, L : \theta_0^{(l,l-1)} \sim \mathcal{N}\left(0, \frac{1}{n_l}\right), \theta_0^{(l)} \sim \mathcal{N}(0, 1) \ . \tag{129}$$

Despite the specificity of this initialization algorithm, it contradicts the broader spirit of this paper. It's not only overly restrictive but also complicates our work by colliding with our framework of tensor's asymptotic behavior. Rather than focusing on a particular initialization scheme like the normal distribution, we will identify and utilize the relevant properties inherent in the distribution.

**Definition E.1** (Appropriate Initialization scheme for Wide Neural Networks). Given a wide neural network as defined above, we characterize the distribution for the initial condition $\theta$ as appropriate if and only if for every probability arbitrarily close to 1, the following properties hold:

1. Different elements of $\theta$ are independent. And for each layer $l = 1, ..., L$, $\theta^{(l,l-1)}$'s and $\theta^{(l)}$'s elements share the same distribution.

2. $\theta$ is symmetric around 0 (implying that all odd moments are nullified):
$$\forall D \in \mathbb{N} \setminus 2\mathbb{N} : \left\langle \theta^{\cdot D} \right\rangle = 0 . \tag{130}$$

3. For every layer $l = 1, ..., L$, all moments of $\theta$ are uniformly normalized:

$$\forall D \in \mathbb{N} : \begin{array}{c} O\left(1\right)^D \leq \frac{1}{\sqrt{n_l}} O\left(\left(\theta^{(l)}\right)^{\cdot D}\right) \leq D! O\left(1\right)^D , \\ O\left(\frac{1}{\sqrt{n_{l-1}}}\right)^D \leq \frac{1}{\sqrt{N_l}} O\left(\left(\theta^{(l,l-1)}\right)^{\cdot D}\right) \leq D! O\left(\frac{1}{\sqrt{n_{l-1}}}\right)^D , \end{array} \quad \text{Uniformly} \tag{131}$$

where $N_l = n_l n_{l-1}$ is the total number of parameters in the $l$-th layer.

where the elemental tensor power defined such as:
$$\forall D \in \mathbb{N} : \left(M^{\cdot D}\right)_{\vec{i}} = M_{\vec{i}}^D . \tag{132}$$

The first two conditions ensure that our system is unbiased, while the third condition guarantees that our system will not be dominated by a disproportionate probabilistic "tail."

We delegate to the reader the verification that Gaussian initialization qualifies as an appropriate initialization.

**Remark E.3.** Conditions (1,2) can be generalized to be fulfilled in the limit of large $n$, provided this convergence occurs rapidly enough. Nevertheless, any complexities arising from this generalization are technical and do not affect our analysis.

**For the remainder of this section, we will omit the biases from our discussion, as they do not add any substantial insights or implications for the points under consideration and won't change any of our results.**

**Lemma E.1** (Normalization of Layers in Proper Wide Neural Networks). Given a wide neural network, if the initial condition is appropriately set, then all the moments across every layer $l = 1...L$ are well normalized:
$$\frac{1}{\sqrt{n_l}} O\left(F^{(l)}\right) = O\left(1\right) . \tag{133}$$

The final parameter that we need to normalize in our system is the dynamic one - the learning rate, denoted by $\eta$. In an attempt to generalize Gaussian initialization, we will adopt the standard method of normalization for $\eta$:
$$\eta \sim \frac{1}{n} . \tag{134}$$

This condition, coupled with the demand for an appropriate initialization strategy, is sufficient to demonstrate that wide neural networks are exponentially weakly correlated PGDML-s.

**In the remainder of this section, we will proceed under the assumption that our parameters are initialized appropriately and that $\eta \sim \frac{1}{n}$.**

We can now use this result to find the asymptotic behavior of the layers derivatives:

**Lemma E.2** (Asymptotic Behavior of Layer's Derivatives). Given our established conditions and initialisation, all derivatives are uniformly bound for each natural number $D$ and layer $l = 1...L$. Specifically, we have:
$$\frac{\eta^{\frac{D}{2}}}{\sqrt{N_D}} O\left(\nabla^{\times D} F^{(l)}\right) \leq O\left(1\right) \quad \text{Uniformly} \tag{135}$$

Here, $N_D = n_l n_{l-1}^D n^D$ represents the asymptotic behavior of the number of elements in the derivatives.

***Proof of lemma (E.1).***

We approach the proof by induction, across the entire proof we use lemma (A.7) to show equivalence between the asymptotic behavior of the system and its tensorial average (67). It is known that the base case, the zeroth layer, naturally satisfies the lemma. By inductive assumption, let us presume that the $l-1$ layer adheres to the lemma. Our task is to establish the lemma's validity for the $l$-th layer for all $l = 1...L$:

$$
\begin{aligned}
\left[F^{(l)}\right]^2 =_1 & \frac{1}{n_l} \sum_i \left\langle \left(\sum_j \theta_{ij}^{(l,l-1)} F_j^{(l-1)}\right) \left(\sum_k \theta_{ik}^{(l,l-1)} F_k^{(l-1)}\right)\right\rangle = \\
& \frac{1}{n_l} \sum_i \left\langle \left(\sum_{j,k} \theta_{ij}^{(l,l-1)} \theta_{ik}^{(l,l-1)} F_j^{(l-1)} F_k^{(l-1)}\right)\right\rangle =_2 \\
& \frac{1}{n_l} \sum_i \sum_{j,k} \left\langle \theta_{ij}^{(l,l-1)} \theta_{ik}^{(l,l-1)}\right\rangle \left\langle F_j^{(l-1)} F_k^{(l-1)}\right\rangle = \\
\frac{1}{n_l} \sum_i \sum_{j \neq k} \left\langle \theta_{ij}^{(l,l-1)} \theta_{ik}^{(l,l-1)}\right\rangle \left\langle F_j^{(l-1)} F_k^{(l-1)}\right\rangle & + \frac{1}{n_l} \sum_i \sum_j \left\langle \left(\theta_{ij}^{(l,l-1)}\right)^2\right\rangle \left\langle \left(F_j^{(l-1)}\right)^2\right\rangle =_3 \\
\sum_i \sum_j \frac{1}{n_l} \left\langle \left(\theta_{ij}^{(l,l-1)}\right)^2\right\rangle \left\langle \left(F_j^{(l-1)}\right)^2\right\rangle =_4 & \sum_{i,j} \frac{1}{n_l} \left\langle \left(\theta_{ij}^{(l,l-1)}\right)^2\right\rangle \sum_k \frac{1}{n_{l-1}} \left\langle \left(F_k^{(l-1)}\right)^2\right\rangle =_5 \\
n_{l-1} \left[\theta^{(l,l-1)}\right]^2 \left[F^{(l-1)}\right]^2 & =_6 O(1) O(1) = O(1) .
\end{aligned}
$$
(136)

Throughout these equalities, we rely on the premise of a proper initialization. Specifically:

- In "1" and "5", we employ the structure of neural networks and the definition of the moment's norm.

- In "2" and "4", we note that $F^{(l-1)}$ is dependent only on the inner parameters of $l$, which are independent of $\theta^{(l,l-1)}$. This is enabled by the proper initialization ensuring $\theta^{(l,l-1)}$ is uniformly distributed.

- In "3", we invoke the fact that different elements of $\theta^{(l,l-1)}$ are independent and symmetric. Hence, for every $i, j \neq k$:

$$
\left\langle \theta_{ij}^{(l,l-1)} \theta_{ik}^{(l,l-1)}\right\rangle = \left\langle \theta_{ij}^{(l,l-1)}\right\rangle \left\langle \theta_{ik}^{(l,l-1)}\right\rangle = 0 .
$$
(137)

- In "6", we apply the induction hypothesis and observe that for a proper initialization (E.1-3):

$$
\forall l = 1...L : \left[\theta^{(l,l-1)}\right] = O\left(\frac{1}{\sqrt{n_{l-1}}}\right) .
$$
(138)

Through the application of the principle of mathematical induction, we conclude the lemma holds for all $l = 1...L$.

Using lemma () again we get that $O\left(F^l\right) \leq O(1)$ but as we know that even if we neglect a small part of the probability distribution the proof should still hold, we get that:

$$
O\left(F^l\right) = O(1) .
$$
(139)

exactly.

$\square$

***Proof of lemma (E.2).*** Given $\omega$, drawn from another proper initialisation, we can observe that $\theta + \omega$ is also properly initialised or sub-properly initialised. Hence, assuming we initialise $F^{(l)}$ accordingly, we find:

$$
\frac{1}{\sqrt{n_l}} O\left(F^{(l)}\right) \leq O(1) .
$$
(140)

Since $F^{(l)}$ is analytical, we can apply its Taylor expansion around $\theta$ to get:

$$
\frac{1}{\sqrt{n_l}} O\left(\sum_{D=0}^{\infty} \nabla^{\times D} F^{(l)}(\theta) \omega^{\times D}\right) \leq O(1) .
$$
(141)

By continuously rescaling $\theta, \omega$ without violating the proper property, we see that all of the component of the expression must be uniformly bounded:

$$\forall D \in \mathbb{N} : \frac{1}{\sqrt{n_l}} O \left( \nabla^{\times D} F^{(l)} (\theta) \omega^{\times D} \right) \leq O(1) \quad \text{Uniformly.} \tag{142}$$

Considering the symmetry of the derivative in its components, and by invoking lemma (A.3), we can identify a vector of size 1 that maximises it, yielding a vector with a size equal to its norm. By setting $\omega$ as this vector and rescaling it to be proper, we obtain using lemma (A.7) that:

$$\forall D \in \mathbb{N} : \frac{1}{\sqrt{n_l}} O \left( \nabla^{\times D} F^{(l)} (\theta) \right) = \frac{1}{\sqrt{n_l}} \frac{1}{\sqrt{n_{l-1}^D}} O \left( \nabla^{\times D} F^{(l)} (\theta) \omega^{\times D} \right) \leq O(1) \quad \text{Uniformly.} \tag{143}$$

Given that:

$$\frac{1}{\sqrt{n_l}} \frac{1}{\sqrt{n_{l-1}^D}} = \frac{1}{\sqrt{n_l n_{l-1}^D}} \sim \frac{\eta^{\frac{D}{2}}}{\sqrt{n_l n_{l-1}^D n^D}} = \frac{\eta^{\frac{D}{2}}}{\sqrt{N_D}} \ , \tag{144}$$

we arrive at the desired result. $\qquad\square$

### E.3   REPRESENTATION OF THE NETWORK'S LAYERS AS A COMPOSITION OF PREVIOUS LAYER COMPONENTS

In this part we use the semilinear structure of wide neural network to establish a linear relation between the correlations of the $l$-th layer to the one of the $l-1$ layer. We will then use this relation next part to show by induction the correlations are weak. For that will define the following useful notation:

**Definition E.2** (Inner and Outer Derivatives). Given a layer $l = 1...L$. We denote the $l$-th layer's outer parameters, which includes its weights (and biases), as follows:

$$\theta_{i^l, i^{l-1}}^{(l, l-1)} \ . \tag{145}$$

Meanwhile, the inner parameters are defined as any of the weights (and biases) from the layers spanning $1...l-1$, and are denoted by:

$$\theta \in \theta^{(-l)} \ . \tag{146}$$

Following the same notation, we denote the gradient of the outer parameters as $\nabla_{(l)}$, and the gradient of the inner parameters as $\nabla_{(-l)}$. The same applies for the correlations, denoted as $\mathfrak{C}_{(l)}, \mathfrak{C}_{(-l)}$.

**Remark E.4.** It is important to note that, as $F^{(l-1)}$ depends only in the inner parameters of the $l$-th layer, the following relationship holds:

$$\nabla_{(-l)} F^{(l-1)} = \nabla F^{(l-1)} \ . \tag{147}$$

This notation can be employed to express the derivative of the $l$-th layer as a combination of derivatives from the $l-1$-th layer.

**Lemma E.3** (Representation of the $l$-th layer derivative, as a combination of its previous layer's derivatives). Given a fully connected wide neural network as specified above, for each $l = 1...L$ layer, the $D \in \mathbb{N}$-th derivative can be presented as follows:

1. When all the derivatives are inner, the expression is:

$$\left( \nabla^{(-l)} \right)^{\times D} F^{(l)} = \theta^{(l, l-1)} \tilde{\nabla}^{\times D} F^{(l-1)} \ . \tag{148}$$

2. When one derivative is outer, and the rest are inner, the expression becomes:

$$\nabla_{i_l i_{l-1}}^{(l)} \times \left( \nabla^{(-l)} \right)^{\times D-1} F_i^{(l)} = \delta_{i i_l} \tilde{\nabla}^{\times D-1} F_{i_{l-1}}^{(l-1)} \ . \tag{149}$$

3. When $2 \leq D$, and for $2 \leq d \in \mathbb{N} \leq D$ where the derivatives are outer, the expression simplifies to:

$$\left(\nabla^{(l)}\right)^{\times d} \times \left(\nabla^{(-l)}\right)^{\times D-d} F^{(l)} = 0 \ . \tag{150}$$

Here, $\tilde{\nabla}^{\times D} F^{(l-1)}$ is the compound derivative, defined such as for $D \in \mathbb{N}$:

$$\tilde{\nabla}^{\times D} F^{(l)} = \sum_{d=1}^{D} \sum_{\substack{d_1 + \ldots + d_d = D \\ d_1 \ldots d_d \in \mathbb{N}}} \phi^{[d]}\left(F^{(l)}\right)\left(\nabla^{\times d_1} F^{(l)} \times \cdots \times \nabla^{\times d_d} F^{(l)}\right) + \text{comb} \tag{151}$$

and for $D = 0$:

$$\tilde{\nabla}_{ij}^{\times 0} F_k^{(l)} = \delta_{ik}\phi\left(F_j\right) \ . \tag{152}$$

The "comb" term refers to all possible combinations of the derivatives' indices. For instance, if we consider one term of the third derivative as follows:

$$\theta^{(l,l-1)}\left(\phi^{[2]}\left(F^{(l-1)}\right)\left(\nabla F^{(l-1)} \times \nabla^{\times 2} F^{(l-1)}\right)\right) \tag{153}$$

then, for every three distinct derivative indices $\alpha_1, \alpha_2, \alpha_3$, there are three unique ways to arrange the indices, disregarding irrelevant parts:

$$\nabla_{\alpha_1} F^{(l-1)} \times \nabla_{\alpha_2 \alpha_3}^{\times 2} F^{(l-1)}, \nabla_{\alpha_2} F^{(l-1)} \times \nabla_{\alpha_1 \alpha_3}^{\times 2} F^{(l-1)}, \nabla_{\alpha_3} F^{(l-1)} \times \nabla_{\alpha_1 \alpha_2}^{\times 2} F^{(l-1)} \ . \tag{154}$$

While the first combination naturally arises from our expression, the "comb" term accounts for the other two.

It should be mentioned that only unique terms are counted, even if they originate from different orders of the derivatives. Therefore, for another component of the third derivative, $\theta^{(l,l-1)}\left(\phi^{[3]}\left(F^{(l-1)}\right)\nabla F^{(l-1)} \times \nabla F^{(l-1)} \times \nabla F^{(l-1)}\right)$, and distinct $\alpha_1, \alpha_2, \alpha_3$:

$$\nabla_{\alpha_1} F^{(l-1)} \nabla_{\alpha_2} F^{(l-1)} \nabla_{\alpha_3} F^{(l-1)}, \nabla_{\alpha_1} F^{(l-1)} \nabla_{\alpha_3} F^{(l-1)} \nabla_{\alpha_2} F^{(l-1)} \ldots \tag{155}$$

are identical, hence should only be counted once.

We can use this result to construct the $l$-th layer correlations using the correlations from the $l-1$ layer:

**Lemma E.4** (Representation of the $l$-th layer correlations, as a combination of its previous layer's correlations). Given the same condition as in (E.3), then:

$$\mathfrak{C}_{(l)}^{D,d} = \theta^{(l,l-1)} \times \left(\tilde{\theta}^{(l,l-1)}\right)^{\times d} \tilde{\mathfrak{C}}_{(l-1)}^{D,d} +$$
$$\eta^{\frac{1}{2}} I \times \eta^{\frac{1}{2}} \phi\left(F^{(l-1)}\right) \times \left(\tilde{\theta}^{(l,l-1)}\right)^{\times d-1} \tilde{\mathfrak{C}}_{(l-1)}^{D,d-1} + \text{comb} + \tag{156}$$
$$\left(\tilde{\theta}^{(l,l-1)}\right)^{\times d} \hat{\mathfrak{C}}_{(l-1)}^{D-1,d} + \text{comb} \ .$$

or when showing the indices explicitly, using Einstein's notation for summation:

$$\left(\mathfrak{C}_{(l)}^{D,d}\right)_{i_0 i_1 \ldots i_d} = \theta_{i_0 j_0}^{(l,l-1)} \tilde{\theta}_{i_1 j_1}^{(l,l-1)} \cdots \tilde{\theta}_{i_d j_d}^{(l,l-1)} \left(\tilde{\mathfrak{C}}_{(l-1)}^{D,d}\right)_{j_0,j_1 \ldots j_d} +$$
$$\eta^{\frac{1}{2}} \delta_{i_0 i_1} \eta^{\frac{1}{2}} \phi\left(F_{j_0}^{(l-1)}\right) \tilde{\theta}_{i_2 j_2}^{(l,l-1)} \cdots \tilde{\theta}_{i_d j_d}^{(l,l-1)} \left(\tilde{\mathfrak{C}}_{(l-1)}^{D,d-1}\right)_{j_0,j_2 \ldots j_d} + \text{comb} + \tag{157}$$
$$\tilde{\theta}_{i_1 j_1}^{(l,l-1)} \cdots \tilde{\theta}_{i_d j_d}^{(l,l-1)} \left(\hat{\mathfrak{C}}_{(l-1)}^{D-1,d}\right)_{i_0,j_1 \ldots j_d} \ ,$$

where the "comb" term includes all index pairings with the zero index, i.e., $(i_0, i_2) \ldots (i_0, i_D)$, and the $\theta$ defined as:

$$\tilde{\theta}_{ij}^{(l,l-1)} = \theta_{ij}^{(l,l-1)} \phi'\left(F_j^{(l-1)}\right) \ . \tag{158}$$

The first compound derivative defined such as for $D \in \mathbb{N}_0, d \in \mathbb{N}$:

$$\tilde{\mathfrak{C}}_{(l)}^{D,d} = \sum_{d'=1}^{D+d} \left\{ C_{\vec{d},\vec{D}} \phi^{[d']}\left(F^{(l)}\right) \mathfrak{C}_{(l)}^{D_1,d_1} \times \cdots \times \mathfrak{C}_{(l)}^{D_{d'},d_{d'}} \ \middle| \ \begin{array}{l} d_1 + \ldots + d_{d'} = d \\ D_1 + \ldots + D_{d'} = D \end{array} \right\} + \text{Comb} \tag{159}$$

where:

$$C_{\vec{d},\vec{D}} = \frac{(D_1! \cdots D_{d'}!)\,(d_1! \cdots d_{d'}!)}{D!d!} \ . \tag{160}$$

Also for $D \in \mathbb{N}_0, d = 0$:

$$\tilde{\mathfrak{C}}_{(l)}^{D,0} = \eta^{\frac{D}{2}} \tilde{\nabla}_t^{\times D} F^{(l)} \ . \tag{161}$$

The second compound derivative defined such as for $D \in \mathbb{N}, d \in \mathbb{N}$:

$$\left(\hat{\mathfrak{C}}_{(l)}^{D-1,d}\right)_{i_0,j_1\ldots j_d}^{\alpha_{d+1}\ldots\alpha_{d+D}} = \eta^{\frac{1}{2}} \delta_{(i_0 j_0)}^{\alpha_{d+1}} \left(\tilde{\mathfrak{C}}_{(l)}^{D-1,d}\right)_{j_0,j_1\ldots j_d}^{\alpha_{d+2}\ldots\alpha_{d+D}} + \text{comb} \ , \tag{162}$$

where the "comb" term is defined as before. For $D = 0$ this compound derivative vanishes.

**Remark E.5.** For the following lemma and the subsequent section, we make the assumption that $D \ll n$. This assumption is permissible even though, in considering the limit, the limit of $D$ should technically be taken prior to that over $n$. This is because higher order derivatives typically exert a decreasing influence over system behavior, leading us to essentially consider them negligible beyond a certain point.

It is important to note that this assumption is not strictly necessary. We could directly address the intricate combinatorial factors without it. Despite this, we prefer to make this assumption to avoid introducing unnecessary complications into our analysis.

**Lemma E.5** (Counting combinations of the derivatives and correlations).

1. For the conditions of lemma (E.3), for every $d_1...d_d$, the number of combinations of the derivatives indices is:
$$\frac{1}{d!} \frac{D!}{d_1! \cdots d_d!} \ , \tag{163}$$
and the total number of combinations above all possible $d = 1...D$-s is the $D$-th "bell number" (which is very close to $D!$).

2. For the conditions of lemma (E.4), for every $d_1...d_{d'}$ and $D_1...D_{d'}$, the number of combinations of the compound correlations is:
$$\frac{1}{d'!} \frac{d!}{d_1! \cdots d_{d'}!} \frac{D!}{D_1! \cdots D_{d'}!} \ . \tag{164}$$

We assume for this lemma the indices are different, as $D \ll n$.

***Proof - lemmas (E.3,E.4).***

We will prove the lemma by induction for a general layer $l = 1...L - 1$ starting with $l = 1$.

The induction base is simple, a this is a direct consequence of taking a derivative over our equation for neural networks (127). This calculation hinges on the concept that, by definition, the inner derivatives are independent of the outer parameters.

$$\nabla_{(l)} F^{(l)} = \nabla_{(l)} \theta^{(l,l-1)} \phi\left(F^{(l-1)}\right) = \theta^{(l,l-1)} \nabla_{(l)} \phi\left(F^{(l-1)}\right) = \\ \theta^{(l,l-1)} \left(\phi^{[1]}\left(F^{(l-1)}\right) \nabla_{(l)} F^{(l-1)}\right) \ , \tag{165}$$

which gives us the induction base.

Assuming by induction our lemma is satisfied for some $D - 1 \in \mathbb{N}$: the inner $D$-th derivative satisfies:

$$\nabla_{(-l)}^{\times D} F^{(l)} = \nabla_{(-l)} \times \nabla_{(-l)}^{\times D-1} F^{(l)} = \\ \nabla_{(-l)} \times \theta^{(l,l-1)} \sum_{d=1}^{D-1} \sum_{d_1\ldots d_d \in \mathbb{N}}^{d_1+\ldots+d_d=D-1} \phi^{[d]}\left(F^{(l-1)}\right) \left(\nabla^{\times d_1} F^{(l-1)} \times \cdots \times \nabla^{\times d_d} F^{(l-1)}\right) \\ + \text{comb} \\ = \\ \theta^{(l,l-1)} \sum_{d=1}^{D-1} \sum_{d_1\ldots d_d \in \mathbb{N}}^{d_1+\ldots+d_d=D-1} \nabla \times \phi^{[d]}\left(F^{(l-1)}\right) \left(\nabla^{\times d_1} F^{(l-1)} \times \cdots \times \nabla^{\times d_d} F^{(l-1)}\right) \\ + \\ \theta^{(l,l-1)} \sum_{d=1}^{D-1} \sum_{d_1\ldots d_d \in \mathbb{N}}^{d_1+\ldots+d_d=D-1} \phi^{[d]}\left(F^{(l-1)}\right) \left(\nabla \times \nabla^{\times d_1} F^{(l-1)} \times \cdots \times \nabla^{\times d_d} F^{(l-1)}\right) \\ + \\ \text{comb}$$

$$\tag{166}$$

We have here a sum of two two different summations, we will analyse each one separably:

Starting from the first one:

$$
\sum_{d=1}^{D-1} \sum_{d_1+\ldots+d_d=D-1 \atop d_1\ldots d_d \in \mathbb{N}} \nabla \times \phi^{[d]} \left( F^{(l-1)} \right) \left( \nabla^{\times d_1} F^{(l-1)} \times \cdots \times \nabla^{\times d_d} F^{(l-1)} \right)
$$
$$
=
$$
$$
\sum_{d=1}^{D-1} \sum_{d_1+\ldots+d_d=D-1 \atop d_1\ldots d_d \in \mathbb{N}} \phi^{[d+1]} \left( F^{(l-1)} \right) \left( \nabla F^{(l-1)} \times \nabla^{\times d_1} F^{(l-1)} \times \cdots \times \nabla^{\times d_d} F^{(l-1)} \right)
$$
$$
=
$$
$$
\sum_{d=1}^{D-1} \sum_{d_1+d_2+\ldots+d_{d+1}=D \atop d_1=1,d_2\ldots d_{d+1} \in \mathbb{N}} \phi^{[d+1]} \left( F^{(l-1)} \right) \left( \nabla^{\times d_1} F^{(l-1)} \times \nabla^{\times d_2} F^{(l-1)} \times \cdots \times \nabla^{\times d_{d+1}} F^{(l-1)} \right)
$$
$$
=
$$
$$
\sum_{d=2}^{D} \sum_{d_1+d_2+\ldots+d_d=D \atop d_1=1,d_2\ldots d_d \in \mathbb{N}} \phi^{[d]} \left( F^{(l-1)} \right) \left( \nabla^{\times d_1} F^{(l-1)} \times \nabla^{\times d_2} F^{(l-1)} \times \cdots \times \nabla^{\times d_d} F^{(l-1)} \right) . \tag{167}
$$

The second term can be represented as:

$$
\sum_{d=1}^{D-1} \sum_{d_1+\ldots+d_d=D-1 \atop d_1\ldots d_d \in \mathbb{N}} \phi^{[d]} \left( F^{(l-1)} \right) \left( \nabla \times \nabla^{\times d_1} F^{(l-1)} \times \cdots \times \nabla^{\times d_d} F^{(l-1)} \right)
$$
$$
=
$$
$$
\sum_{d=1}^{D-1} \sum_{(d_1+1)+\ldots+d_d=D \atop d_1\ldots d_d \in \mathbb{N}} \phi^{[d]} \left( F^{(l-1)} \right) \left( \nabla^{\times d_1+1} F^{(l-1)} \times \cdots \times \nabla^{\times d_d} F^{(l-1)} \right) \tag{168}
$$
$$
=
$$
$$
\sum_{d=1}^{D-1} \sum_{d_1+\ldots+d_d=D \atop 1<d_1 \in \mathbb{N},d_2\ldots d_d \in \mathbb{N}} \phi^{[d]} \left( F^{(l-1)} \right) \left( \nabla^{\times d_1} F^{(l-1)} \times \cdots \times \nabla^{\times d_d} F^{(l-1)} \right) .
$$

Combining the two sums we get exactly the form that we were searching for, which finishes the proof of the lemma's first case.

Lemma (E.4) is a direct result. $\qquad\square$

***Proof - lemma (E.5).***

Proving the lemma's first part:

The number of way to sort into $d$ distinct sets with $d_1...d_d$ objects is:

$$
\frac{(d_1 + \cdots d_d)!}{d_1! \cdots d_d!} = \frac{D!}{d_1! \cdots d_d!} , \tag{169}
$$

but our sets are not distinct, so we need to divide by the appropriate coefficient. But if the sets are not the same, they repeat in different arrangements, so we get the $\frac{1}{d!}$. summing over all of these options we get the definition of the $D$-th bell number.

The second part is the same. $\qquad\square$

### E.4  WIDE FNC-S ARE WEAKLY CORRELATED PGDML SYSTEMS

Here we will show a detailed heuristic proof of why wide neural networks are weakly correlated PGDML as described in (E.2).

**Remark E.6.** For this section we assume that the width of the last layer, i.e the $L$-th layer is exactly $L = 1$. That won't impact any of our results of the system asymptotic behavior as $L$ is fixed in $n$ as discussed in remark (A.6).

**Remark E.7.** In the entire section we will use Einstein's summation notation (liberally).

We initiate our exploration of wide neural network correlations (and derivatives) by focusing on the most critical one - the kernel - $\mathfrak{C}^1$.

For the final layer $l = L$, the kernel norm is simply expressed as:

$$
\left\| \mathfrak{C}_{(L)} \right\| = \left| \mathfrak{C}_{(L)} \right| . \tag{170}
$$

Given that $n_L = 1$, the kernel is merely a scalar.

Leveraging lemma (E.4), we can construct the $L$-th layer kernel from the components of the preceding layer:

$$
\mathfrak{C}_{(L)}^1 = \theta_i^{(L,L-1)} \theta_j^{(L,L-1)} \left( \mathfrak{C}_{(L-1)}^1 \right)_{ij} + \eta \phi \left( F_j^{(L-1)} \right)^2 . \tag{171}
$$

Applying lemma (E.1) and the Lipschitz property of $\phi$, we discern that the right term has the asymptotic behavior of $\eta\phi\left(F_j\right)^2 \sim O\left(1\right)$. Concerning the left term, lemma (E.4) once again provides:

$$\left(\mathfrak{C}^1_{(L-1)}\right)_{ij} = \theta^{(L,L-1)}_{ip}\theta^{(L,L-1)}_{jq}\left(\mathfrak{C}^1_{(L-2)}\right)_{pq} + \delta_{ij}\eta\phi\left(F^{(L-2)}_k\right)^2 . \tag{172}$$

This means we have an $O(1)$ term and another that depends on the previous term. Continuing this process by induction and employing the fact that everything is symmetric, hence positive, we conclude that the kernel's asymptotic behavior is precisely $O(1)$. **In combination with (E.2), we find that our system satisfies the criteria of a PGDML (B.2)!**

Let's now consider a general $D \in \mathbb{N}_0, d \in \mathbb{N}$ final correlation. By invoking lemma (A.3), we know that there exists a vector $v \in S_N$ achieving the norm:

$$\left\|\mathfrak{C}^{D,d}_{(L)}\right\| = \left|\mathfrak{C}^{D,d}_{(L)} \cdot v^{\times D}\right| . \tag{173}$$

Applying lemma (E.4), we find that this expression can be constructed from $D-1$ correlations. Considering only the first term among the three in the equation, (the treatment for others would be the same), and focusing solely on the first correlations, we obtain (up to $\frac{1}{d!}$ when omitting the $\frac{1}{D!}$ as we do not consider the different combinations):

$$\left(\phi^{[d+D]}\left(F^{(L-1)}\right)\left(\theta^{(L,L-1)}\right)\right) \times \left(\tilde{\theta}^{(L,L-1)}\right)^{\times d} \cdot \left(\left(\mathfrak{C}_{(L-1)}\right)^{\times d} \times \left(\eta^{\frac{1}{2}}\nabla F^{(L-1)}\right)^{\times D} \cdot v^{\times D}\right) . \tag{174}$$

Using (23), and that the $L-1$ layer and $L$ are independent at initialization, we can dismiss the $\phi$-s, leaving the asymptotic behavior unchanged (we would discuss the $d!$ later):

$$\left(\theta^{(L,L-1)}\right)^{\times d+1} \cdot \left(\left(\mathfrak{C}_{(L-1)}\right)^{\times d} \times \left(\eta^{\frac{1}{2}}\nabla F^{(L-1)}\right)^{\times D} \cdot v^{\times D}\right) . \tag{175}$$

When constructing the kernels from the preceding layer, as each one consists of two terms (171), resulting in $2^d$ terms in total. This factor of $2^d$ does not alter the system's asymptotic behavior, so instead, we can consider only the maximal terms, which are the ones with only one kind of first correlation terms. We will choose the first kind of terms, dealing with the others via induction:

$$\theta^{(L,L-1)}_{i_0}\theta^{(L,L-1)}_{i_1}\cdots\theta^{(L,L-1)}_{i_d}\left(\delta_{i_0 i_1}\eta\phi\left(F^{(L-2)}_k\right)^2\right)\cdots\left(\delta_{i_0 i_d}\eta\phi\left(F^{(L-2)}_k\right)^2\right) \cdot$$
$$\left(\left(\eta^{\frac{1}{2}}\nabla F^{(L-2)}_{i_0}\right)^{\times D} \cdot v^{\times D}\right) . \tag{176}$$

As $\eta\phi\left(F^{(L-2)}_k\right)^2 \sim O(1)$, after reducing the deltas, we obtain an asymptotic behavior of at most:

$$\left(\theta^{(L,L-1)}_i\right)^{d+1}\left(\left(\eta^{\frac{1}{2}}\nabla F^{(L-2)}_{i_0}\right)^{\times D} \cdot v^{\times D}\right) . \tag{177}$$

Now, as we already have that $O\left(\eta^{\frac{1}{2}}\nabla F^{(L-2)}_{i_0}\right) \leq O\left(1\right)$, if $D \in \mathbb{N}$ we find multiplied by a vector of at most size $O(1)$. In the worst case, this object will have an asymptotic behavior of:

$$\left(\theta^{(L,L-1)}_i\right)^{d+2} . \tag{178}$$

We know from our proper initialization that it is uniformly bounded for all $d$-s by:

$$d!O\left(\frac{1}{\sqrt{n}}\right)^d . \tag{179}$$

which means that by reintroducing the $\frac{1}{d!}$ we get:

$$O\left(\frac{1}{\sqrt{n}}\right)^d . \tag{180}$$

If $D = 0$ however, the $\left( \eta^{\frac{1}{2}} \nabla F_{i_0}^{(L-2)} \right)^{\times D} \cdot v^{\times D}$ term disappears and we are left with:

$$\left( \theta_i^{(L,L-1)} \right)^{d+1} . \tag{181}$$

For odd $d$-s, we still have $O\left( \frac{1}{\sqrt{n}} \right)^d$ as $\theta$ is symmetric. However, for even ones, we find:

$$O\left( \frac{1}{\sqrt{n}} \right)^{d-1} . \tag{182}$$

**This explains why, while our system is $n$ weakly correlated, it is only $n^{\frac{2}{3}}$ power weakly correlated.** Nonetheless, for the time deviation, one can easily confirm that this term remains negligible as $n \to \infty$.

Of course, there are many other terms rather then the first derivatives ones. But we can they can be treated similarly.

Assuming that for $l - 1$ layer:

$$\phi^{[d']} \left( F^{(l-1)} \right) \mathfrak{C}_{(l-1)}^{D_1,d_1} \times \cdots \times \mathfrak{C}_{(l-1)}^{D_{d'},d_{d'}} . \tag{183}$$

contributes at most:

$$O\left( \frac{1}{\sqrt{n}} \right)^{d \text{ or } d-1} . \tag{184}$$

We get utilizing Lemma (E.5), and replacing $\phi^{[d']} \left( F^{(l-1)} \right) \to d'!$ (as warranted by equation (23)), we find that the total contribution is bounded by:

$$\sum_{d'=1}^{D+d} \sum \sum \frac{1}{d'!} \frac{d!}{d_1! \cdots d_{d'}!} \frac{D!}{D_1! \cdots D_{d'}!} d'! \frac{d_1! \cdots d_{d'}!}{d!} \frac{D_1! \cdots D_{d'}!}{D!} O\left( \frac{1}{\sqrt{n}} \right)^{d \text{ or } d-1}$$
$$\sim 2^{D+d} O\left( \frac{1}{\sqrt{n}} \right)^{d \text{ or } d-1} \sim O\left( \frac{1}{\sqrt{n}} \right)^{d \text{ or } d-1} . \tag{185}$$

In a similar vein, it can be demonstrated that multiple correlations taken together exhibit the same behavior at the $l$-th layer. Which means that we can prove by induction in the same way we did for the first correlations, that all of them behave the same, **thereby concluding our (heuristic) proof**.

### E.5 GENERALISATION BEYOND FNC-S

#### E.5.1 TENSOR PROGRAMS

While FNCs are the prototypical network architecture, numerous other architectures are utilized practice as we discussed in section (4.2). The tensor programs formalism, as detailed in (Yang & Littwin (2021)), offers a unified language to encapsulate most relevant neural network architectures, by viewing them as a composites of global linear operations and pointwise nonlinear functions. This formalism encompasses an extensive array of neural network architectures, including recurrent neural networks and attention-based networks. In their work they demonstrated that any wide network described by this formalism exhibit linearization.

Our weak correlation approach naturally aligns with the tensor programs framework, simplifying the proof that such networks not only exhibit linearization, but also are low correlated PGDMLs. This comes with all of the additional implications that, like deviations over learning and the influence of network augmentation on the linearization rate.

Our proof for FNCs can be simply generalised for any wide network described by this formalism, because, similarly to FNCs, all such systems exhibit a wide semi-linear form by definition.

#### E.5.2 BEYOND TENSOR PROGRAMS

Given the broad generality of the tensor programs formalism, it's challenging to devise linearizing networks that fall outside its scope. However, here we suggest two network-based architectures that demonstrate linearization and, to our belief, stand outside this formalism.

The first is FNC as outlined in equation (127), but where each neuron possesses a unique activation function:

$$F_i^{(l)} = \sum_{j=1}^{n_{l-1}} \theta_{ij}^{(l,l-1)} \phi_j \left( F_j^{(l-1)} \right) + \theta_i^{(l)} . \tag{186}$$

The proof of the linearization of this system, assuming $\phi_i$ satisfies condition (23), simply parallels our proof for FNCs.

Not all such systems are outside the random tensor formalism's purview, if we can represent $\phi_i$ as a function of two distinct inputs - $F_i$ and another external input given by the index $j \in \mathbb{N}$, such as:

$$\forall j = 1...n_{l-1} : \phi_j \left( F_j^{(l-1)} \right) = \phi \left( F_j^{(l-1)}, j \right) . \tag{187}$$

However, since $\phi$ and all its derivatives must remain bounded by some polynomial to fit within the theorems of Yang & Littwin (2021); Yang (2020) for wide neural networks, if $\phi_i$ is exceedingly diverse, pinpointing a suitable $\phi$ could be very challenging or even impossible.

A more definite (albeit synthetic) example of a linearizing network-based system outside the tensor programs realm can be formulated as:

$$z(x) = \sum_{i=1}^{n} \theta_i f_i(x) + \sum_{i,j=1}^{n} \theta_i \theta_j g_i(x) g_j(x) \quad g = Af , \tag{188}$$

initialized by $\theta = 0$, where $A$ is a $90°$ rotating matrix across the relevant axis as $n \to \infty$, and $f_i$ are chosen as the eigenfunctions of some external kernel.

This system can be viewed just an NTK approximation, but with a non-trivial second derivative that is perpendicular to the first. Hence, our system will still behave linearly as $n \to \infty$. It's also not evident how this system can be derived from the tensor programs framework.

While one might contend that this example seems artificially contrived to the point of limiting its significance, it underscores the existence of weakly correlated, network-based systems that are not encapsulated by the tensor programs formalism.

Furthermore, in line with our discourse in (3.3.3), if we manage to discern the types of effective correlations that could prove advantageous, such systems might find practical applications.

## F  LIMITATIONS, FURTHER DISSECTION AND GENERALISATION

In this section, we enumerate the key assumptions that underpin our analysis and propose potential extensions to our findings beyond these stipulated preconditions. Additionally, we identify potential avenues for related further research.

In this section, we enumerate the key assumptions that underpin our analysis and propose potential extensions to our findings beyond these stipulated preconditions. Additionally, we identify potential avenues for related further research.

### F.1  SECTION (2)

Our analysis here did not rely on any hidden or nontrivial assumptions, except for those explicitly stated during the tensor definition. Our findings are generalizable and applicable to any random tensor or variable that is dependent on some limiting parameter $n \in \mathbb{N}$. Extending our results to any set with a total order is straightforward.

We anticipate this analytical tool to be beneficial not only for the investigation of wide neural networks but also for the learning of random tensors and variables in general, particularly when focusing on their limiting behavior, for reasons delineated in this paper. It upholds several useful algebraic properties (A.3), provides a well-defined, optimal asymptotic bound for any tensor (2.1), and harmonizes naturally with the notion of "convergence in distribution". Further, owing to its inherent generality, it offers widespread applicability. We recommend further exploration into the utilization of this tool in solving other problems.

### F.2 SECTIONS (3,4)

#### F.2.1 ASSUMPTIONS

1. We presuppose that $F$, $\mathcal{C}$, and $\phi$ are analytical in their parameters, that is, they are smooth and their Taylor series converge to them.

2. All of $\phi$ derivatives are bounded such as in equation (23).

3. Our analysis is constrained to the case of single-batch stochastic gradient descent, and we assume that our training and testing distributions coincide.

4. We presume $\mathcal{C}$ is convex that is, $\mathcal{C}''$ is a positivly defined.

5. Our theorems (3.1,3.2) and corollary (4.0.1) are exclusively applicable to PGDML systems, as defined in (B.2).

6. Theorem (3.1) and corollary (4.0.1) are valid only for sufficiently small $\eta$ that is of the same order of magnitude as the $\eta$ necessary for effective linear studies.

7. Corollary (4.0.1) stipulates that the first derivative of $\mathcal{C}$ decays exponentially, and the second derivative remains bounded over time for the linear solution.

8. The equivalence showed in theorems 3.1,3.2 demand that all of the derivatives stay fixed. But one can describe a more nuanced equivalence, where the derivatives do significantly change, but the network itself do behaves linearly, if this change is perpendicular to $\nabla F(\theta_0)$. However, given the fact that neural networks satisfy our simpler conditions we will remain with the above stated version of the equivalence.

#### F.2.2 GENERALIZATIONS OF THE ASSUMPTIONS

For condition (1), while we typically deal with smooth analytical functions, non-continuous hypothesis functions are common, as with the "ReLU" activation function in neural networks. If our system can be represented as a linear approximation plus a function that is analytical over patches, with the understanding that non-smooth points are of zero measure, then the techniques presented herein can be applied.

Regarding the bound imposed on the derivatives of $\phi$, (2), this bound is relatively non-restrictive. Especially considering that $\phi$ should be analytic and this condition only needs to hold over an arbitrarily large probability set, not the entire probability space.

Extending the single-input batch gradient descent case (3) to other batch schemes, such as multiple-input batches or deterministic single batch GD, is straightforward. This extension simply involves replicating our work while adjusting the specifics of the optimization algorithm of interest. The generalization for more complex gradient-based algorithms follows similar lines, albeit with more nuances.