# OpenReview forum: "Weak Correlations as the Underlying Principle for Linearization of Gradient-Based Learning Systems"
_ICLR.cc/2024/Conference — Submitted to ICLR 2024_

### Official Review · Reviewer_VTRJ · 2023-10-24

**Soundness:** 1 poor
**Presentation:** 1 poor
**Contribution:** 4 excellent
**Rating:** 1
**Confidence:** 4

**Summary:**

This paper tries to establish weak correlations among derivatives of an NN wrt to its degrees of freedom as the underlying mechanism of linearization of gradient-based learning behaviour in infinitely or very wide neural networks. More precisely, it is claimed that (i) NNs exhibit weak correlations, and (ii) gradient-based (possibly more general) learning systems linearize under weak correlations.
To this end, the author(s) introduce the concept of random tensor asymptotic behaviour in order to establish bounds on said derivative correlations, and in such a manner that the Neural Tangent Kernel exhibits a special case. It is then claimed a tight bound on higher order derivative correlations, eventually leading to the traditional NTK regime, i.e. that vanishing higher order contributions effectively constitute the training linearization (and not the other way around).

**Strengths:**

(1) The paper pursues a highly ambitious and highly relevant question, providing possibly groundbreaking insights. E.g. if the claims show to be true, this would ensue further research on how correlations can be induced in large NNs in order to overcome linearization and/or the loss of representation learning in wide deep learning, generally on how to leverage correlations in order to control training dynamics.

(2) The paper introduces the framework of random tensor asymptotic behaviour, which promises to facilitate several other applications in the field

**Weaknesses:**

As much as I appreciate the importance of the research question and the mathematical elaborateness, the paper exhibits the following problems:

(1) Clarity / Presentation is immature: While the posed claims of the paper and the introduced methodological notions (NTK, random tensor asymptotic behaviour) are crystal clear, the presentation of logical reasoning and evidence to support the claims is rather hard to follow. Even after studying the supplementary material, the overall synthesis of the many involved steps, aspects and "directions" is incoherent/confusing.  Most notably, "m(n)" is used to establish the main theorems, but the definition of "m(n)" is nowhere to be found (also not in the supplementary), except for a prosaic description of "the typical parameter of the linearization/correlation decay where m(n) → ∞". This is in contradiction to the seeming mathematical rigour and elaborateness.

(2) The paper presents no experiment(s). The work would strongly benefit from numerical examples to more clearly support the claims and illustrate the implications & significance of this work. This should at least include 2 manufactured toy examples, where (i) linearization is demonstrated in correspondence with weak correlations, as well as (ii) the opposite of that. Also the transition regime is interesting. Or even better, maybe even real-world applications can be found for demonstration

**Questions:**

Some suggestions have been given above, more minor questions/comments:

(1) The presentation would certainly benefit from (i) a coherent reorganization, (ii) replacing in statements like "as we see/explore/demonstrate later" the word "later" with a reference to a concrete section of the paper, (iii) supporting numerous statements that are "evident" at least with references, and (iv) a complete introduction of the non-standart notation (several symbols are not properly introduced like \phi or \Delta, and sometimes I had to guess the meaning of notation, e.g. \nabla^{\times d} or the big-O subscripts), maybe even streamlining the notation. E.g. the meaning of the crucial limiting parameter is discussed only several pages after theorems have been stated with it. For building intuition with the reader, it could be helpful to first discuss a simple low-rank example

(2) The paper would benefit from a clear delimitation of scope, discussion of limitations, disadvantages

(3) The authors state "Our theorem will be applicable solely for systems that are properly scaled in the
initial condition where n → ∞". What does that mean?

(4) There are several incomplete sentences

---

> ### Author Response · Authors · 2023-11-22
> **Thank you for your important review, comments and suggestions.**
>
> Thank you for your detailed and insightful review. We have attached an updated version of our paper, which we believe significantly enhances its content. We also aim to refine it further. Our responses for your review have been integrated into section 6 in the new version of the supplementary material (to be removed in the camera-ready version) to facilitate easier referencing of specific aspects of our work and related literature.

---

> > ### Comment · Reviewer_VTRJ · 2023-11-22
> > **comments not implemented, revisions unclear**
> >
> > A quick look tells me my major point 2 (missing experiments) was not considered (contrary to what is stated in the author response? also, no section 6 in the supplementary). Major point 1 I cannot judge so quickly.
> > As for the rebuttal, a point-by-point response would be necessary beyond a generic response. As for the revision, a color markup of changes would be helpful.

---

### Official Review · Reviewer_YXgU · 2023-10-26

**Soundness:** 2 fair
**Presentation:** 2 fair
**Contribution:** 2 fair
**Rating:** 3
**Confidence:** 3

**Summary:**

The problem it aims is to give precise criteria for linearized training (i.e. NTK-like) to occur (e.g. we know it occurs in the limit of infinite width given a specific parameterization [Jacot et al. 2018] or when we add a large scaling factor [Chizat et al. 2020]). It also claims to prove that wide NNs satisfy these precise criteria but this result is not properly stated in the main.

A half of the paper is devoted to a formalism on measuring asymptotic behavior of random tensors. This formalism aims to give precise definition for "M_n = O_{n \to \infty} (a_n)", where M_n is a sequence of random tensors and a_n is a number sequence. This definition has nothing to do with the tensor itself but rather deals directly with its operator norm. Therefore it should be applicable to sequences of random variables; it would be helpful for the reader to understand how this notion is different to the usual "stochastic Big O".

The results are presented in a very general form making them difficult to consume. I would suggest the authors putting a simplified formulation in the main, maybe also a proof sketch, as well as some application examples, in particular *emphasizing the cases where their analysis allows one to gain insight over the existing methods*.

The significance of the results are not convincing. When are these results able to prove linearized training in cases where [Lee et al., 2019] does not apply? Both results are asymptotic in nature; what are the cases when the claimed results are stronger?

The paper contains no experimental evidence that the results are applicable in practice, it would be good to add some.

The literature review is a little thin. The paper does not mention the work of [Dyer & Gur-Ari, 2020] that seems very relevant.

**Strengths:**

The overall problem is interesting, and the local structure of the paper is reasonable (no typos, definitions are always provided, etc.). Any substantial contribution to understanding when linearization occurs or doesn't would be very valuable (though the abstract formulation makes it a little hard to see in the present form).

**Weaknesses:**

The key weakness that needs to be addressed before I can recommend this for publication is a clear presentation of what the method gives and doesn't give, in particular in comparison to existing methods (and a discussion of how this improves on the existing literature).

The paper also takes of tangents with notation and results that cannot be particularly relevant to the main goal. For instance, there are double factorials in the main, but they appear nowhere in the final result (and again, given that it is a 'soft' result, they cannot be of any importance for the whole result).

Also, some things are not correctly written (e.g. the NTK does not converge to the target function as written somewhere).

**Questions:**

Can you give a clear intuition of what we learn about neural networks? Is your approach a conceptualization of earlier approaches or a different novel idea? How is one expect to use your result?

---

> ### Author Response · Authors · 2023-11-22
> **Thank you for your valuable review and suggestions for improving the paper.**
>
> Thank you for your detailed and insightful review. We have attached an updated version of our paper, which we believe significantly enhances its content. We also aim to refine it further. Our responses for your review have been integrated into section 6 in the new version of the supplementary material (to be removed in the camera-ready version) to facilitate easier referencing of specific aspects of our work and related literature.

---

### Official Review · Reviewer_ZGhT · 2023-10-31

**Soundness:** 2 fair
**Presentation:** 1 poor
**Contribution:** 1 poor
**Rating:** 3
**Confidence:** 4

**Summary:**

This paper asks why overparameterized neural networks can be linearised with respect to their parameters (e.g. in the Neural Tangent Kernel regime), and propose that the reason is weak correlations between the first and higher derivatives of the model function. With respect to previous work, they consider the case of neural networks with two distinct activation functions and the deviation from linearity during SGD training.

**Strengths:**

Understanding the behaviour of overparameterized neural networks is a very important and interesting question.

**Weaknesses:**

In my opinion this work fails on providing and/or communicating anything new on the topic.

1) Discussion on some very important related work is missing, which this work should have compare with.
2) Several statements are unsupported, definitions are missing, there are several inaccuracies and the paper is overall very hard to follow.
3) The mathematical notation is cumbersome and, for no apparent reason, completely different from many related papers.
4) Crucial points are relegated to the appendix, without which the main text is severely incomplete.

The main reference missing is “On the linearity of large non-linear models: when and why the tangent kernel is constant”, NeurIPS 2020 by Liu, Zhu, Belkin (https://arxiv.org/abs/2010.01092), but there are many other papers following this one that have studied the question of why neural networks can be linearised, also in relation to the model derivatives.
This line of work is not discussed at all.

i) As a main contribution, the author list the case of “wide neural networks with two distinct activation functions”, but the only thing they say about this case in the main text is one sentence on page 9, relegating all about this claim in the appendix (we are not even told what “neural networks with two distinct activation functions” mean in the main text).
ii) Section 2 is completely unmotivated and its relevance remains unclear until much later. For example, in the first three paragraphs of section 2.2 it’s unclear what is the goal and the challenges in reaching the goal.
iii) The function \Epsilon is supposed to be a generic convex function, but there seems to be an (unstated) assumption that it depends on the difference between F and y, which is true for the square loss but not for many other commonly used loss functions.
iv) (x,y) are called, respectively, label and images, but it should be the other way around.
v) A “limiting parameter” in introduced on page 5 but never explained in the main text
vi) Equation 9 only applies to gradient flow, not to gradient descent. After reading Theorem 3.1 it becomes clear that Equation 9 is a definition, but until then it just looks like a mistake.
vii) Below Equation 12, Why can the parameter indices viewed as random variables? They are not random variables, and they are not drawn from a uniform distribution. Instead, all indices are summed over all the parameters. If they were a sample from a uniform distribution, there would be some noise.
viii) No intuition is given here about the relevance of the quantity introduced in equation 12.
ix) I don't understand Definition 3.2. "O" is supposed to be limiting order. What is "O" there?
x) n0 not defined in equations (16) and (17)
xi) There should not be a Delta in the second expression of equation (17)
xii) Inequality 19 seems to be crucial for obtaining the results. However, the statement "nearly all realistic scalable systems satisfy" is not justified.
xiii) What does "typical parameter of the linearization/correlation decay" mean?
xiv) “This scenario is a little more complex but can be dealt with.” How is this scenario dealt with? It seems the reader here just needs to trust the authors without any explanation or justification.
xv) “These systems can be interpreted as non-linear dynamical systems.” Any reference for this statement?

i) The Jacobian of the model is called “”derivative matrix and is transposed with respect to the Jacobian that everyone uses.
ii) I have never seen a gradient with a subscript “T” to denote the transpose of the operation result.
iii) In equation 12, the gradient with several subscripts and superscripts is just a (high order) partial derivative. Why re-inventing the notation?

**Questions:**

NA

---

> ### Author Response · Authors · 2023-11-22
> **Thank you for your detailed and important review. It is very instructive.**
>
> Thank you for your detailed and insightful review. We have attached an updated version of our paper, which we believe significantly enhances its content. We also aim to refine it further. Our responses for your review have been integrated into section 6 in the new version of the supplementary material (to be removed in the camera-ready version) to facilitate easier referencing of specific aspects of our work and related literature.

---

### Meta-Review · Area_Chair_ByuG · 2023-12-13

**Metareview:**

This paper studies the mechanism that the linearization of neural network occurs. They showed that the weak correlation between the first and second order derivatives induce the linearization.

Unfortunately, the writing of this paper is not good. The organization of the paper can be much improved, and there are several mathematically non-rigorous statements and missing definitions even in the main statement. Then, it is hard to see the main contribution exactly. Moreover, several related work are not properly cited. Then, I cannot recommend acceptance for this paper.

**Justification For Why Not Higher Score:**

The writing of this paper is clearly under the bar for ICLR. The novelty and technical difficulty of the main statement are not well clarified. Therefore, it is difficult to recommend "Acceptance."

**Justification For Why Not Lower Score:**

N/A

---

### Decision · Program_Chairs · 2024-01-16

Reject